# Non-Abelian braiding of graph vertices in a superconducting processor

Google Quantum AI and Collaborators*

Indistinguishability of particles is a fundamental principle of quantum mechanics[1]. For all elementary and quasiparticles observed to date—including fermions, bosons and Abelian anyons—this principle guarantees that the braiding of identical particles leaves the system unchanged[2,3]. However, in two spatial dimensions, an intriguing possibility exists: braiding of non-Abelian anyons causes rotations in a space of topologically degenerate wavefunctions[4–8]. Hence, it can change the observables of the system without violating the principle of indistinguishability. Despite the well-developed mathematical description of non-Abelian anyons and numerous theoretical proposals[9–22], the experimental observation of their exchange statistics has remained elusive for decades. Controllable many-body quantum states generated on quantum processors offer another path for exploring these fundamental phenomena. Whereas efforts on conventional solid-state platforms typically involve Hamiltonian dynamics of quasiparticles, superconducting quantum processors allow for directly manipulating the many-body wavefunction by means of unitary gates. Building on predictions that stabilizer codes can host projective non-Abelian Ising anyons[9,10], we implement a generalized stabilizer code and unitary protocol[23] to create and braid them. This allows us to experimentally verify the fusion rules of the anyons and braid them to realize their statistics. We then study the prospect of using the anyons for quantum computation and use braiding to create an entangled state of anyons encoding three logical qubits. Our work provides new insights about non-Abelian braiding and, through the future inclusion of error correction to achieve topological protection, could open a path towards fault-tolerant quantum computing.

Elementary particles in three dimensions are either bosons or fermions. The existence of only two types is rooted in the fact that the worldlines of two particles in three plus one dimensions can always be untied in a trivial manner. Hence, exchanging a pair of indistinguishable particles twice is topologically equivalent to not exchanging them at all, and the wavefunction must remain the same. Representing the exchange as a matrix $R$ acting on the space of wavefunctions with a constant number of particles, it is thus required that $R^2 = 1$ (a scalar), leaving two possibilities: $R = 1$ (bosons) and $R = -1$ (fermions). Such continuous deformation is not possible in two dimensions, thus allowing collective excitations (quasiparticles) to show richer braiding behaviour. In particular, this permits the existence of Abelian anyons[2,3,6–8,24,25], in which the global phase change due to braiding can take any value. It has been proposed that there exists another class of quasiparticles known as non-Abelian anyons, in which braiding instead results in a change of the observables of the wavefunction[4,5,24]. In other words, $R^2$ does not simplify to a scalar, but remains a unitary matrix. Therefore, $R^2$ is a fundamental characteristic of anyon braiding. The topological approach to quantum computation[26] aims to leverage these non-Abelian anyons and their topological nature to enable gate operations that are protected against local perturbations and decoherence errors[5,27–30]. In solid-state systems, primary candidates of non-Abelian quasiparticles are low-energy excitations in Hamiltonian systems, including the 5/2 fractional quantum Hall states[31,32], vortices in topological superconductors[33,34] and Majorana zero modes in semiconductors proximitized by superconductors[35–38]. However, direct verification of non-Abelian exchange statistics has remained elusive[39–41].

We formulate the necessary requirements for experimentally certifying a physical system as a platform for topological quantum computation[5,26]: (1) create an anyon pair; (2) verify the rules that govern the 'collision' of two anyons, known as the fusion rules; (3) verify the non-Abelian braiding statistics reflected in the matrix structure $R^2$ and (4) realize controlled entanglement of anyonic degrees of freedom. Notably, the observation of steps (2)–(4) requires measurements of multi-anyon states, by means of fusion or non-local measurements.

The advent of quantum processors allows for controlled unitary evolution and direct access to the wavefunction rather than the parameters of the Hamiltonian. These features enable the use of local operations for efficient preparation of topological states that can host non-Abelian anyons, and—as we will demonstrate—their subsequent braiding and fusion. Moreover, these platforms allow for probing arbitrary Pauli strings through destructive multiqubit (that is, non-local) measurements. As the braiding of non-Abelian anyons in this platform is achieved through unitary gate control rather than adiabatic evolution of a Hamiltonian system, we note that the anyons are not quasiparticles

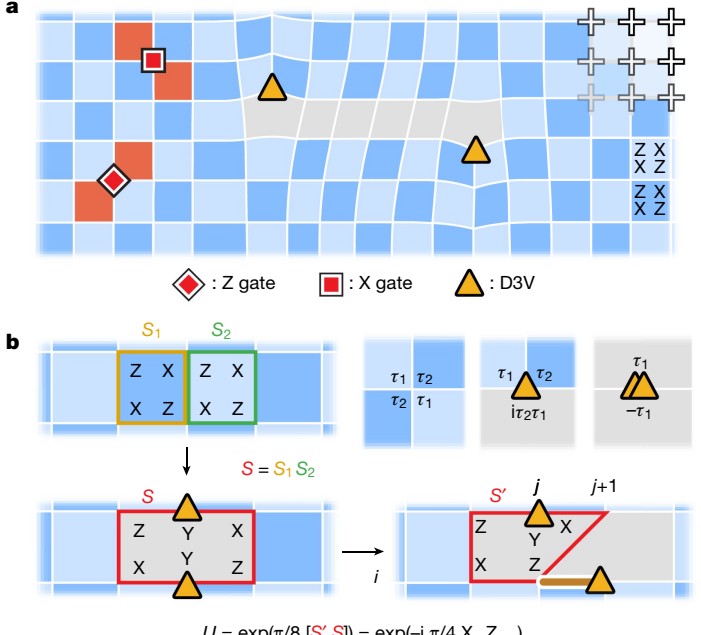

**a**

◇ : Z gate   ▧ : X gate   ▲ : D3V

**b**

$U = \exp(\pi/8 \, [S',S]) = \exp(-i\,\pi/4\, X_{i,j}\, Z_{i,j+1})$

**Fig. 1 | Deformations of the surface code. a**, Stabilizer codes are conveniently described in a graph framework. Through deformations of the surface code graph, a square grid of qubits (crosses) can be used to realize more generalized graphs. Plaquette violations (red) correspond to stabilizers with $s_p = -1$ and are created by local Pauli operations. In the absence of deformations, plaquette violations are constrained to move on one of the two sublattices of the dual graph in the surface code, hence the two shades of blue. **b**, A pair of D3Vs in the sense of eigenstates that persist throughout a Hamiltonian evolution. (yellow triangles) appears by removing an edge between two neighbouring stabilizers, $\hat{S}_1$ and $\hat{S}_2$, and introducing the new stabilizer, $\hat{S} = \hat{S}_1\hat{S}_2$. A D3V is moved by applying a two-qubit entangling gate, $\exp\!\left(\frac{\pi}{8}[\hat{S}', \hat{S}]\right)$. In the presence of bulk D3Vs, there is no consistent way of chequerboard colouring, hence the (arbitrarily chosen) grey regions. The top right shows that in a general stabilizer graph, $\hat{S}_p$ can be found from a constraint at each vertex, where $\{\tau_1, \tau_2\} = 0$.

in the sense of eigenstates that persist throughout a Hamiltonian evolution. Their movement is achieved through local operations along their paths, and they are kept spatially separated throughout the braiding. We therefore emphasize that the two-dimensional braiding processes are physically taking place on the device, leading to actual non-Abelian exchange effects of local anyons in the many-body wavefunction, rather than matrix operations that simply follow the same algebra.

To realize a many-body quantum state that can host anyons, it is essential to control the topological degeneracy. A suitable platform for achieving this requirement is a stabilizer code[42], in which the wavefunctions are characterized by a set of commuting operators $\{\hat{S}_p\}$ called stabilizers, with $\hat{S}_p |\psi\rangle = s_p |\psi\rangle$ and $s_p = \pm 1$. The code space is the set of degenerate wavefunctions for which $s_p = 1$ for all $p$. Hence, every independent stabilizer divides the degeneracy of the code space by two.

Whereas the physical layout of qubits is typically used to determine the structure of the stabilizers, the qubits can be considered to be degree $j$ vertices (D$j$V; $j \in \{2, 3, 4\}$) on more general planar graphs (Fig. 1a)[23]. Using this picture, each stabilizer can be associated with a plaquette $p$, whose vertices are the qubits on which $\hat{S}_p$ acts:

$$\hat{S}_p = \prod_{v \,\in\, \text{vertices}} \hat{\tau}_{p,v}. \tag{1}$$

$\hat{\tau}_{p,v}$ is here a single-qubit Pauli operator acting on vertex $v$, chosen to satisfy a constraint around that vertex (Fig. 1b). An instance where $s_p = -1$ on a plaquette is called a plaquette violation. These can be thought of as quasiparticles, which are created and moved through single-qubit Pauli operators (Fig. 1a). A pair of plaquette violations sharing an edge constitute a fermion, $\varepsilon$. We recently demonstrated the Abelian statistics of such quasiparticles in the surface code[43]. To realize non-Abelian statistics, one needs to go beyond such plaquette violations; it has been proposed that dislocations in the stabilizer graph—analogous to lattice defects in crystalline solids—can host projective

non-Abelian Ising anyons[9,10]. For brevity, we refer to these as 'non-Abelian anyons' or simply 'anyons' from here on.

In the graph framework introduced above, it has been shown that such dislocations are characterized as vertices of degree 2 and 3 (ref. 23). Consider the stabilizer graph of the surface code[26,44], specifically with boundary conditions such that the degeneracy is two. Although all the vertices in the bulk are D4Vs, one can create two D3Vs by removing an edge between two neighbouring plaquettes $p$ and $q$, and introducing the new stabilizer $\hat{S} = \hat{S}_p\hat{S}_q$ (Fig. 1b). Evidently, the introduction of two D3Vs reduces the number of independent stabilizers by one and thus doubles the degeneracy. This doubling is exactly what is expected when a pair of Ising anyons is introduced[9,10]; hence, D3Vs appear as a candidate of non-Abelian anyons, and we will denote them as $\sigma$.

To be braided and fused by unitary operations, the D3Vs must be moved. Whereas the structure of the stabilizer graph is usually considered to be static, it was predicted by Bombin that the dislocations in the surface code would show projective non-Abelian Ising statistics if braided[10]. Here, we will use a specific protocol recently proposed by Lensky et al.[23] for deforming the stabilizer graph (and thus moving the anyons) using local two-qubit Clifford gates. To shift a D3V from vertex $u$ to $v$, an edge must be disconnected from $v$ and reconnected to $u$. This can be achieved by means of the gate unitary $\exp\!\left(\frac{\pi}{8}[\hat{S}'_p, \hat{S}_p]\right)$, where $\hat{S}_p$ is the original stabilizer containing the edge and $u$, and $\hat{S}'_p$ is the new stabilizer that emerges after moving the edge[23]. In cases where the D3V is shifted between two connected vertices, the unitary simplifies to the form $U_{\pm}(\hat{\tau}_u\hat{\tau}_v) \equiv \exp\!\left(\pm i\frac{\pi}{4}\hat{\tau}_u\hat{\tau}_v\right)$, where $\hat{\tau}_u$ and $\hat{\tau}_v$ are Pauli operators acting on vertices $u$ and $v$. We experimentally realize this unitary through a controlled Z (CZ) gate and single-qubit rotations (median errors of $7.3 \times 10^{-3}$ and $1.3 \times 10^{-3}$, respectively; Methods).

Following these insights from Kitaev and Bombin, we now turn to our experimental study of the proposed anyons, using the protocol described in ref. 23. In the first experiment, we demonstrate the creation of anyons and the fundamental fusion rules of $\sigma$ and $\varepsilon$ (Fig. 2a).

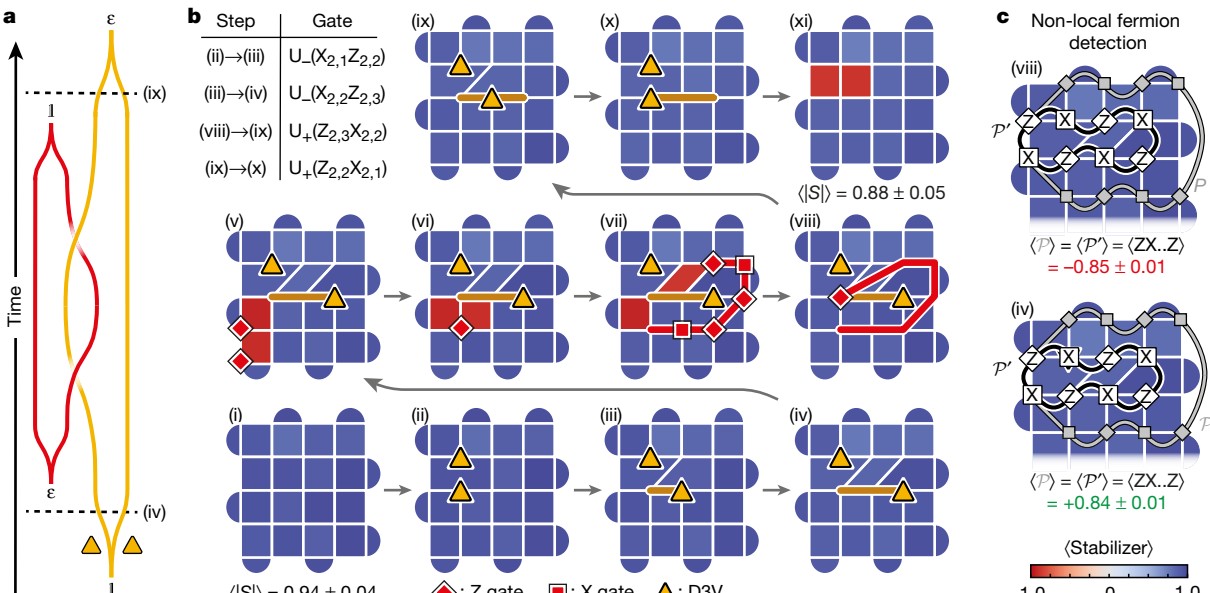

**Fig. 2 | Demonstration of the fundamental fusion rules of D3Vs. a**, The braiding worldlines used to fuse $\varepsilon$ and $\sigma$. **b**, Expectation values of stabilizers at each step of the unitary operation after readout correction (see Extended Data Fig. 3 for details and individual stabilizer values). We first prepare the ground state of the surface code (step (i); average stabilizer value of $0.94 \pm 0.04$, where the uncertainty is one standard deviation). A D3V ($\sigma$) pair is then created (ii) and separated (iii)–(iv), before creating a fermion, $\varepsilon$ (v). One of the plaquette violations is brought around the right $\sigma$ (vi)–(viii), allowing it to annihilate with the other plaquette violation (viii). The fermion has seemingly disappeared, but re-emerges when the $\sigma$ are annihilated ((xi); stabilizer values $-0.86$ and $-0.87$). The path (v) → (viii) demonstrates the fusion rule, $\sigma \times \varepsilon = \sigma$. The different fermion parities at the end of the paths (viii) → (xi) and (iv) → (i) show the other fusion rule, $\sigma \times \sigma = \mathbb{1} + \varepsilon$. Yellow triangles represent the positions of the $\sigma$. The brown and red lines denote the paths of the $\sigma$ and the plaquette violation, respectively. Red squares (diamonds) represent X (Z) gates. Upper left shows a table of two-qubit unitaries used in the protocol. Each stabilizer was measured $n = 10,000$ times in each step. **c**, A non-local technique for hidden fermion detection: the presence of a fermion in a $\sigma$-pair can be deduced by measuring the sign of the Pauli string $\hat{\mathcal{P}}$ corresponding to bringing a plaquette violation around the $\sigma$-pair (grey path). $\hat{\mathcal{P}}$ is equivalent to the shorter string $\hat{\mathcal{P}}'$ (black path). Measurements of $\hat{\mathcal{P}}'$ in steps (viii) (top) and (iv) (bottom) give values of $-0.85 \pm 0.01$ and $+0.84 \pm 0.01$, respectively. This indicates that there is a hidden fermion pair in the former case, but not in the latter, despite the stabilizers being the same.

In a $5 \times 5$ grid of superconducting qubits, we first use a protocol consisting of four layers of CZ gates to prepare the surface code ground state (Fig. 2b(i), see also ref. 43). The average stabilizer value after the ground state preparation is $0.94 \pm 0.04$ (individual stabilizer values shown in Extended Data Fig. 3c). We then remove a stabilizer edge to create a pair of D3Vs ($\sigma$) and separate them through the application of two-qubit gates. Fig. 2b(i)–(iv) show the measured stabilizer values in the resultant graph in each step of this procedure (determined by simultaneously measuring the involved qubits in their respective bases, $n = 10,000$; note that the measurements are destructive and the protocol is restarted after each measurement). In Fig. 2b(v), single-qubit Z gates are applied to two qubits near the lower left corner of the grid to create adjacent plaquette violations, which together form a fermion. Through the sequential application of X and Z gates (Fig. 2b(vii)–(viii)), one of the plaquette violations is then made to encircle the right $\sigma$ vertex. Crucially, after moving around $\sigma$, the plaquette violation does not return to where it started, but rather to the location of the other plaquette violation. This enables them to annihilate (Fig. 2b(viii)), causing the fermion to seemingly disappear. However, by bringing the two $\sigma$ back together and annihilating them (Fig. 2b(ix)–(xi)), we arrive at a striking observation: an $\varepsilon$ particle re-emerges on two of the square plaquettes where the $\sigma$ vertices previously resided.

Our results demonstrate the fusion of $\varepsilon$ and $\sigma$. The disappearance of the fermion from step (v) to (viii) establishes the fundamental fusion rule of $\varepsilon$ and $\sigma$:

$$\sigma \times \varepsilon = \sigma. \tag{2}$$

We emphasize that none of the single-qubit gates along the path of the plaquette violation are applied to the qubits hosting the mobile $\sigma$;

our observations are therefore solely due to the non-local effects of non-Abelian D3Vs, and exemplify the unconventional behaviour of the latter. Moreover, another fusion rule is seen by considering the reverse path (iv) → (i), and comparing it to the path (viii) → (xi). These two paths demonstrate that a pair of $\sigma$ can fuse to form either vacuum ($\mathbb{1}$) or one fermion (steps (i) and (xi), respectively):

$$\sigma \times \sigma = \mathbb{1} + \varepsilon. \tag{3}$$

The starting points of these two paths ((iv) and (viii)) cannot be distinguished by any local measurement. We therefore introduce a non-local measurement technique that allows for detecting an $\varepsilon$ without fusing the $\sigma$ (refs. 10,23,26). The key idea underlying this method is that bringing a plaquette violation around a fermion should result in a $\pi$ phase. We therefore measure the Pauli string $\hat{\mathcal{P}}$ that corresponds to creating two plaquette violations, bringing one of them around the two $\sigma$, and finally annihilating them with each other (grey paths in Fig. 2c). The existence of an $\varepsilon$ inside the $\sigma$-pair should cause $\langle \hat{\mathcal{P}} \rangle = -1$. To simplify this technique further, $\hat{\mathcal{P}}$ can be reduced to a shorter string $\hat{\mathcal{P}}'$ (black paths in Fig. 2c) by taking advantage of the stabilizers it encompasses. For instance, if $\hat{\mathcal{P}}$ contains three of the operators in a four-qubit stabilizer, these can be switched out with the remaining operator. Measuring $\langle \hat{\mathcal{P}}' \rangle$ in step (iv), in which the $\sigma$ are separated but the fermion has not yet been introduced, gives $\langle \hat{\mathcal{P}}' \rangle = +0.84 \pm 0.01$, consistent with the absence of fermions (Fig. 2c). However, performing the exact same measurement in step (viii), in which the $\sigma$ are in the same positions, we find $\langle \hat{\mathcal{P}}' \rangle = -0.85 \pm 0.01$, indicating that an $\varepsilon$ is delocalized across the spatially separated $\sigma$ pair (Fig. 2c). This observation highlights the non-local encoding of the fermions, which cannot be explained with classical physics.

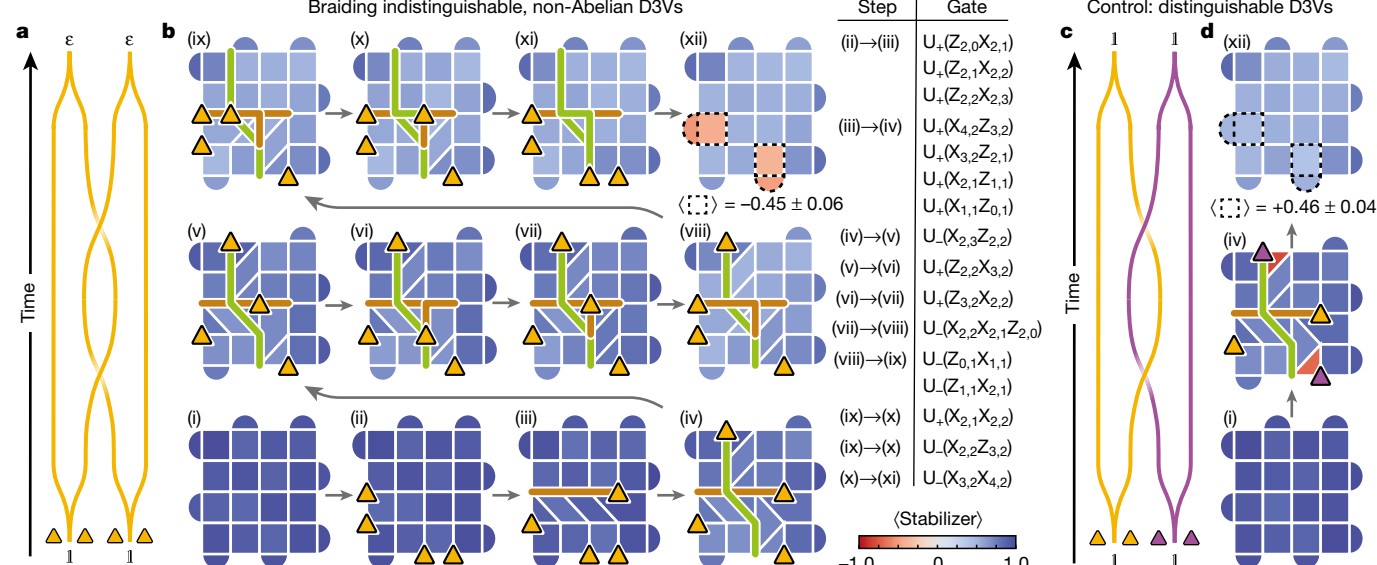

**Fig. 3 | Braiding of non-Abelian anyons. a**, Wordline schematic of the braiding process. **b**, Experimental demonstration of braiding, showing the values of the stabilizers throughout the process. Two $\sigma$ pairs, A and B, are created from the vacuum $\mathbb{1}$, and one of the $\sigma$ in pair A is brought to the right side of the grid. Next, a $\sigma$ from pair B is moved to the top, thus crossing the path of pair A, before bringing $\sigma$ pairs A and B back together to complete the braid. In the final step, two fermions appear in the locations where the $\sigma$ pairs resided, constituting a change in the local observables. The diagonal $\sigma$ move in step (iv) requires two SWAP gates (three CZ gates each) and a total of ten CZ gates. The three-qubit unitary in step (viii) requires four SWAP gates and a total of 15 CZ gates. In the full circuit, a total of 40 layers of CZ gates are applied (Methods). The yellow triangles represent the locations of the $\sigma$; the brown and green lines represent the paths of $\sigma$ from pairs A and B, respectively. The four red stabilizers in (xii) have a mean value of $-0.45 \pm 0.06$, where the uncertainty is one standard deviation. Each stabilizer was measured $n = 10,000$ times in each step. **c**, As a control experiment, we perform the same braid as in **a**, but with distinguishable $\sigma$ by attaching a plaquette violation to the $\sigma$ in pair B (represented with purple triangles). **d**, Same as **b**, but using distinguishable $\sigma$ (only showing steps (i), (iv) and (xii)). In contrast to **b**, no fermions are observed in step (xii).

Having demonstrated the above fusion rules involving $\sigma$, we next braid them with each other to directly show their non-Abelian statistics. We consider two spatially separated $\sigma$ pairs, A and B, by removing two stabilizer edges (Fig. 3a,b(ii)). Next, we apply two-qubit gates along a horizontal path to separate the $\sigma$ in pair A (Fig. 3b(iii)), followed by a similar procedure in the vertical direction on pair B (Fig. 3b(iv)), so that one of its $\sigma$ crosses the path of pair A. We then subsequently bring the $\sigma$ from pairs A and B back to their original positions (Fig. 3b(v)–(viii) and (ix)–(xi), respectively). When the two $\sigma$ pairs are annihilated in the final step (Fig. 3b(xii)), we observe that a fermion is revealed in each of the positions where the $\sigma$ pairs resided (average stabilizer value $-0.45 \pm 0.06$). This shows a clear change in local observables from the initial state in which no fermions were present. As a control experiment, we repeat the experiment with distinguishable $\sigma$ pairs, achieved by attaching a plaquette violation to each of the $\sigma$ in pair B (Fig. 3c,d; see also Extended Data Fig. 8 for stabilizer measurements through the full protocol). Moving the plaquette violation along with the $\sigma$ requires a string of single-qubit gates, which switches the direction of the rotation in the multiqubit unitaries, $U_{\pm} \to U_{\mp}$. In this case, no fermions are observed at the end of the protocol (average stabilizer value $+0.46 \pm 0.04$), thus providing a successful control.

Fermions can only be created in pairs in the bulk. Moreover, the fusion of two $\sigma$ can only create zero or one fermion (equation (3)). Hence, our experiment involves the minimal number of bulk $\sigma$ (four) needed to encode two fermions and demonstrate non-Abelian braiding. Because the fermion parity is conserved, effects of gate imperfections and decoherence can be partially mitigated by postselecting for an even number of fermions. This results in fermion detection values of $-0.76 \pm 0.03$ and $+0.79 \pm 0.04$ in Fig. 3b,d, respectively.

Together, our observations show the change in local observables by braiding of indistinguishable $\sigma$ and constitute a direct demonstration of their non-Abelian statistics. In other words, the double-braiding operation $R^2$ is a matrix that cannot be reduced to a scalar. Specifically, it corresponds to an $X$ gate acting on the space spanned by zero- and two-fermion wavefunctions.

The full braiding circuit consists of 40 layers of CZ gates and 41 layers of single-qubit gates (36 of each after ground state preparation). The effects of imperfections in this hardware implementation can be assessed through comparison with the control experiment. The absolute values of the stabilizers in which the fermions are detected in the two experiments (dashed boxes in Fig. 3b,d(xii)) are very similar (average values of $-0.45$ and $+0.46$). This is consistent with the depolarization channel model, in which the measured stabilizer values are proportional to the ideal values ($\pm 1$).

We next study the prospects of using D3Vs to encode logical qubits and prepare an entangled state of anyon pairs. By doubling the degeneracy, each additional $\sigma$ pair introduces one logical qubit, where the $|0\rangle_L$ ($|1\rangle_L$) state corresponds to an even (odd) number of hidden fermions. The measurements of the fermion numbers in several $\sigma$ pairs are not fully independent: bringing a plaquette violation around one $\sigma$ pair is equivalent to bringing it around all the other pairs (due to the conservation of fermionic parity). Hence, $N \geq 2$ anyons encode $N/2 - 1$ logical qubits. The D3Vs we have created and manipulated so far are not the only ones present in the stabilizer graph; with the boundary conditions used here, each of the four corners are also D3Vs, no different from those in the bulk[23]. Indeed, the existence of D3Vs in the corners is the reason why a single fermion could be created in the corner in Fig. 2b(v). This is also consistent with the fact that the surface code itself encodes one logical qubit in the absence of additional D3Vs. Here we create two pairs of D3Vs, in addition to the four that are already present in the corners, to encode a total of three logical qubits.

Through the use of braiding, we aim to prepare an entangled state of these logical qubits, specifically a GHZ state on the form $(|000\rangle + |111\rangle)/\sqrt{2}$. The definition of a GHZ state and the specifics of how it is prepared is basis-dependent. In most systems, the degrees of

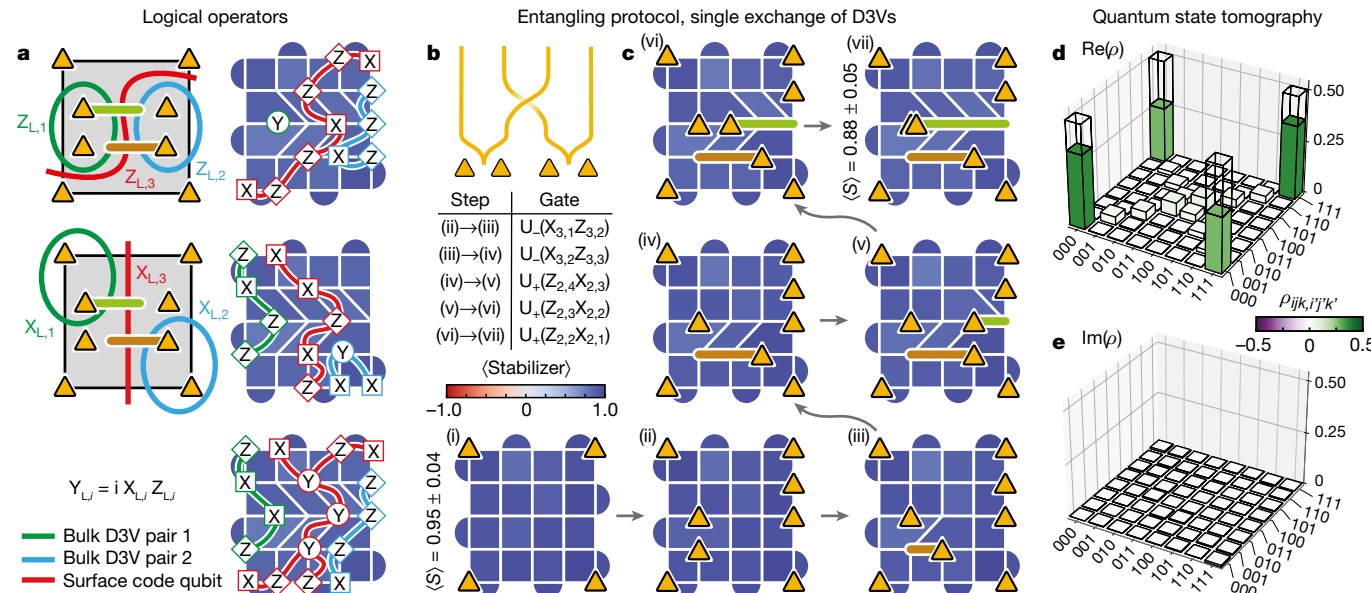

**Fig. 4 | Entangled state of anyon-encoded logical qubits by means of braiding. a**, Logical operators of the three logical qubits encoded in the eight anyons (other basis choices are possible). The coloured curves in the left column denote plaquette violation paths, before reduction to shorter, equivalent Pauli strings measured in the experiment (right column). **b**, Worldline schematic of the single exchange used to realize an entangled state of the logical qubits. **c**, Single exchange of the non-Abelian anyons, showing measurements of the stabilizers throughout the protocol. Yellow triangles represent the locations of

the σ, whereas brown and green lines denote their paths. The average stabilizer values are 0.95 ± 0.04 and 0.88 ± 0.05 (one standard deviation) in the first and last step, respectively. Each stabilizer was measured $n = 20,000$ times in each step. **d,e**, Real (**d**) and imaginary (**e**) parts of the reconstructed density matrix from the quantum state tomography. Re($\rho$) has clear peaks in its corners, as expected for a GHZ state on the form $(|000\rangle + |111\rangle)/\sqrt{2}$. The overlap with the ideal GHZ state is Tr$\{\rho_{\mathrm{GHZ}}\rho\} = 0.623 \pm 0.004$, where the uncertainty is one standard deviation determined from bootstrapping.

freedom are local and there is a natural choice of basis. For spatially separated anyons, the measurement operators are necessarily non-local. Here we choose the basis defined as follows: for the first two logical qubits, we choose the logical $\hat{Z}_{\mathrm{L},i}$ operators to be Pauli strings encircling each of the bulk σ pairs, as was used in Fig. 2c (green and turquoise paths in the left column of Fig. 4a). For the logical surface code qubit, we define $\hat{Z}_{\mathrm{L},3}$ as the Pauli string that crosses the grid horizontally through the gap between the bulk D3V pairs, effectively enclosing four σ (red path in Fig. 4a). In this basis, the initial state is a product state.

Whereas a double braid was used to implement the operator $X$ in Fig. 3, we now perform a single braid (Fig. 4b) to realize $\sqrt{X}$ and create a GHZ state. We implement this protocol by bringing one σ from each bulk pair across the grid to the other side (Fig. 4c). For every anyon double exchange across a Pauli string, the value of the Pauli string changes sign. Hence, a double exchange would change $|000\rangle$ to $|111\rangle$, whereas a single exchange is expected to realize the superposition, $(|111\rangle + |000\rangle)/\sqrt{2}$.

To study the effect of this operation, we perform quantum state tomography on the final state, which requires measurements of not only $\hat{Z}_{\mathrm{L},i}$, but also $\hat{X}_{\mathrm{L},i}$ and $\hat{Y}_{\mathrm{L},i}$ on the three logical qubits. For the first two logical qubits, $\hat{X}_{\mathrm{L},i}$ is the Pauli string that corresponds to bringing a plaquette violation around only one of the σ in the pair (as demonstrated in Fig. 2b). Both the logical $\hat{X}_{\mathrm{L},i}$ and $\hat{Z}_{\mathrm{L},i}$ operators can be simplified by reducing the original Pauli strings (green and turquoise paths in the left column of Fig. 4c) to equivalent, shorter ones (right column). $\hat{Z}_{\mathrm{L},1}$ can in fact be reduced to a single $\hat{Y}$-operator. For the logical surface code qubit, we define $\hat{X}_{\mathrm{L},3}$ as the Pauli string that crosses the grid vertically between the bulk D3V pairs (red path in Fig. 4a). Finally, the logical $\hat{Y}_{\mathrm{L},i}$-operators are simply found from $\hat{Y}_{\mathrm{L},i} = i\hat{X}_{\mathrm{L},i}\hat{Z}_{\mathrm{L},i}$. Measuring these operators, we reconstruct the density matrix of the final state (Fig. 4d,e), which has a purity of $\sqrt{\mathrm{Tr}\{\rho^2\}} = 0.646 \pm 0.003$ and an overlap with the ideal GHZ state of Tr$\{\rho_{\mathrm{GHZ}}\rho\} = 0.623 \pm 0.004$ (uncertainties estimated

from bootstrapping method; resampled 10,000 times from the original data set). The fact that the state fidelity is similar to the purity suggests that the infidelity is well described by a depolarizing error channel.

In conclusion, we have realized highly controllable braiding of degree-3 vertices, enabling the demonstration of the fusion and braiding rules of non-Abelian Ising anyons. We have also shown that braiding can be used to create an entangled state of three logical qubits encoded in these anyons. In other, more conventional candidate platforms for non-Abelian exchange statistics, which involve Hamiltonian dynamics of quasi-particle excitations, topological protection naturally arises from an emergent gap that separates the computational states from other states. To leverage the non-Abelian anyons in our system for topologically protected quantum computing, the stabilizers must be measured throughout the braiding protocol. The potential inclusion of this error correction procedure, which involves overheads including readout of five-qubit stabilizers, could open a new path towards fault-tolerant implementation of Clifford gates, a key ingredient of universal quantum computation.

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

**Google Quantum AI and Collaborators**

T. I. Andersen[1], Y. D. Lensky[2], K. Kechedzhi[1], I. K. Drozdov[1,3], A. Bengtsson[1], S. Hong[1], A. Morvan[1], X. Mi[1], A. Opremcak[1], R. Acharya[1], R. Allen[1], M. Ansmann[1], F. Arute[1], K. Arya[1], A. Asfaw[1], J. Atalaya[1], R. Babbush[1], D. Bacon[1], J. C. Bardin[1,4], G. Bortoli[1], A. Bourassa[1], J. Bovaird[1], L. Brill[1], M. Broughton[1], B. B. Buckley[1], D. A. Buell[1], T. Burger[1], B. Burkett[1], N. Bushnell[1], Z. Chen[1], B. Chiaro[1], D. Chik[1], C. Chou[1], J. Cogan[1], R. Collins[1], P. Conner[1], W. Courtney[1], A. L. Crook[1], B. Curtin[1], D. M. Debroy[1], A. Del Toro Barba[1], S. Demura[1], A. Dunsworth[1], D. Eppens[1], C. Erickson[1], L. Faoro[1], E. Farhi[1], R. Fatemi[1], V. S. Ferreira[1], L. F. Burgos[1], E. Forati[1], A. G. Fowler[1], B. Foxen[1], W. Giang[1], C. Gidney[1], D. Gilboa[1], M. Giustina[1], R. Gosula[1], A. G. Dau[1], J. A. Gross[1], S. Habegger[1], M. C. Hamilton[1,5], M. Hansen[1], M. P. Harrigan[1], S. D. Harrington[1], P. Heu[1], J. Hilton[1], M. R. Hoffmann[1], T. Huang[1], A. Huff[1], W. J. Huggins[1], L. B. Ioffe[1], S. V. Isakov[1], J. Iveland[1], E. Jeffrey[1], Z. Jiang[1], C. Jones[1], P. Juhas[1], D. Kafri[1], T. Khattar[1], M. Khezri[1], M. Kieferová[1,6], S. Kim[1], A. Kitaev[1], P. V. Klimov[1], A. R. Klots[1], A. N. Korotkov[1,7], F. Kostritsa[1], J. M. Kreikebaum[1], D. Landhuis[1], P. Laptev[1], K.-M. Lau[1], L. Laws[1], J. Lee[1,8], K. W. Lee[1], B. J. Lester[1], A. T. Lill[1], W. Liu[1], A. Locharla[1], E. Lucero[1], F. D. Malone[1], O. Martin[1], J. R. McClean[1], T. McCourt[1], M. McEwen[1,9], K. C. Miao[1], A. Mieszala[1], M. Mohseni[1], S. Montazeri[1], E. Mount[1], R. Movassagh[1], W. Mruczkiewicz[1], O. Naaman[1], M. Neeley[1], C. Neill[1], A. Nersisyan[1], M. Newman[1], J. H. Ng[1], A. Nguyen[1], M. Nguyen[1], M. Y. Niu[1], T. E. O'Brien[1], S. Omonije[1], A. Petukhov[1], R. Potter[1], L. P. Pryadko[1,10], C. Quintana[1], C. Rocque[1], N. C. Rubin[1], N. Saei[1], D. Sank[1], K. Sankaragomathi[1], K. J. Satzinger[1], H. F. Schurkus[1], C. Schuster[1], M. J. Shearn[1], A. Shorter[1], N. Shutty[1], V. Shvarts[1], J. Skruzny[1], W. C. Smith[1], R. Somma[1], G. Sterling[1], D. Strain[1], M. Szalay[1], A. Torres[1], G. Vidal[1], B. Villalonga[1], C. V. Heidweiller[1], T. White[1], B. W. K. Woo[1], C. Xing[1], Z. J. Yao[1], P. Yeh[1], J. Yoo[1], G. Young[1], A. Zalcman[1], Y. Zhang[1], N. Zhu[1], N. Zobrist[1], H. Neven[1], S. Boixo[1], A. Megrant[1], J. Kelly[1], Y. Chen[1], V. Smelyanskiy[1], E.-A. Kim[2,11,12,13 ✉], I. Aleiner[1 ✉] & P. Roushan[1 ✉]

[1]Google Research, Mountain View, CA, USA. [2]Department of Physics, Cornell University, Ithaca, NY, USA. [3]Department of Physics, University of Connecticut, Storrs, CT, USA. [4]Department of Electrical and Computer Engineering, University of Massachusetts, Amherst, MA, USA. [5]Department of Electrical and Computer Engineering, Auburn University, Auburn, AL, USA. [6]QSI, Faculty of Engineering & Information Technology, University of Technology Sydney, Sydney, New South Wales, Australia. [7]Department of Electrical and Computer Engineering, University of California, Riverside, CA, USA. [8]Department of Chemistry, Columbia University, New York, NY, USA. [9]Department of Physics, University of California, Santa Barbara, CA, USA. [10]Department of Physics and Astronomy, University of California, Riverside, CA, USA. [11]Department of Physics, Ewha Womans University, Seoul, South Korea. [12]Department of Physics, Harvard University, Cambridge, MA, USA. [13]Radcliffe Institute for Advanced Studies, Cambridge, MA, USA.

## Methods

### Qubit decoherence and gate characterization

The experiments are performed on a quantum processor with frequency-tuneable transmon qubits and a similar design to that in ref. 45. Extended Data Fig. 1a shows the measured relaxation times of the 25 qubits that were used in the experiment, with a median value of $T_1 = 21.7$ µs. We also measure the dephasing time $T_2$ in a Hahn echo experiment, shown in Extended Data Fig. 1b, with the same median value of 21.7 µs. We note that the equality of T1 and T2 is a coincidence and that the discrepancy between the measured decoherence rate $1/T_2$ and the relaxation-limited rate $1/(2T_1)$ is due to remnant noise not decoupled in the Hahn echo experiment.

Next, we benchmark the gates used in the experiment. Extended Data Fig. 2a,b show the cumulative distribution of the Pauli errors for single- and two-qubit (CZ) gates, respectively. The median Pauli errors are $1.3 \times 10^{-3}$ for the single-qubit gates and $7.3 \times 10^{-3}$ for the two-qubit gates.

### Readout details

Because the readout of the qubit state is imperfect, the raw data gives a somewhat incorrect representation of the actual quantum state of the system. We write the probability of readout error of state 0(1) on qubit $i$ as $p_{0(1),i}$, and the readout fidelity is thus given by $1 - (p_{0,i} + p_{1,i})/2$. To correct for any asymmetry between readout of the $|0\rangle$ and $|1\rangle$ states, we perform symmetrized measurements in which $\pi$ pulses are applied to the qubits before the readout in half of the measurements and the recorded qubit values are inverted. The measured value of a stabilizer with actual value $\langle S \rangle = \langle \prod_i \alpha_i \rangle$ (where the product runs over qubits in the stabilizer) is then:

$$\langle S \rangle_{\text{meas}} = \langle \prod_i (1 - p_{0,i} - p_{1,i}) \alpha_i \rangle = \prod_i (1 - p_{0,i} - p_{1,i}) \langle S \rangle, \quad (4)$$

where we made use of the fact that each qubit is measured equally often in the $|0\rangle$ and $|1\rangle$ states in the symmetrized measurements. Note the absence of the factor 1/2 compared to the expression for the readout fidelity, as perfectly incorrect readout ($p_0 = p_1 = 1$) would give a readout fidelity of 0, but a measured value of $-\alpha_i$. To correct for the discrepancy between the measured stabilizer value and the actual stabilizer value, we measure $\langle Z_1 . . . Z_n \rangle$ of the state $|00 .. 00\rangle$ with the same qubits (using again symmetrized measurements) to find:

$$\langle Z_1 . . Z_n \rangle_{\text{meas}} = \prod_i 1 - p_{0,i} - p_{1,i} \quad (5)$$

The readout-corrected $\langle S \rangle$ is then found from:

$$\langle S \rangle_{\text{corr}} = \langle S \rangle_{\text{meas}} / \langle Z_1 . . Z_n \rangle_{\text{meas}} \quad (6)$$

Extended Data Fig. 3 shows the measured readout errors, as well as a comparison of the stabilizer values in the surface code ground state (same data as Fig. 2b(i)) before and after readout correction.

### Dynamical decoupling

To mitigate the effects of qubit decoherence during the circuits, we perform dynamical decoupling on qubits that are idle for more than three layers of gates. In particular, we use the Carr–Purcell–Meiboom–Gill scheme, consisting of X pulses interspaced by a wait time of $\tau = 25$ ns. Extended Data Fig. 4 shows an example comparison of the stabilizers in cases with and without dynamical decoupling, after braiding of anyons (41 layers of SQ gates and 40 layers of CZ gates). A clear improvement is observed, increasing the average absolute stabilizer value from 0.50 to 0.58.

### Circuit details

Extended Data Fig. 5 shows the circuits used in the experiments presented in the main text. In our experiment, the two-qubit unitaries $U_\pm(\hat{t}_1 \hat{t}_2)$ are converted to single-qubit rotations and CZ gates, as shown in Extended Data Fig. 6b. In the particular case in which a D3V is moved diagonally (Fig. 3b(iv)), we realize the unitary by including two SWAP gates (also converted to CZ gates) as the qubits are connected in a square grid (Extended Data Fig. 6c). Moreover, the three-qubit unitary in Fig. 3b(viii) is equivalent to a combination of single-qubit gates, four SWAP gates and four CZ gates (Extended Data Fig. 6d), which can be further converted to single-qubit gates and 15 CZ gates. In the experimental implementation of the circuit, adjacent single-qubit gates on the same qubit are merged and performed in the layer after the most recent CZ gate (Extended Data Fig. 6e).

### Numerical simulation of braiding in presence of noise

To better understand the role of errors in the experimental results in Fig. 3 of the main text, we perform a numerical simulation of the density matrix evolution subject to the braiding circuit in the presence of noise. We use the method of quantum trajectories to approximate the expectation value of stabilizers with the 25-qubit density matrix. The model of noise includes $T_1$ and $T_2$ effects described by the single-qubit Kraus operators,

$$K_0 = \begin{pmatrix} 1 & 0 \\ 0 & \exp(-t/T_2) \end{pmatrix}, \quad (7)$$

$$K_1 = \begin{pmatrix} 0 & \sqrt{1 - \exp(-t/T_1)} \\ 0 & 0 \end{pmatrix}, \quad (8)$$

$$K_2 = \begin{pmatrix} 0 & 0 \\ 0 & \sqrt{\exp(-t/T_1) - \exp(-2t/T_2)} \end{pmatrix}, \quad (9)$$

where $t$ is the duration of the evolution, as well as additional one- and two-qubit depolarizing channel error for each gate. The depolarizing channel error rate is chosen such that the combined Pauli error from $T_1$, $T_2$ and depolarizing error matches the gate Pauli error measured in an independent experiment (Qubit decoherence and gate characterization). We take these values to be uniform across the chip. The expectation values of the four stabilizers that correspond to the noise-free value of −1, see light red stabilizers in Fig. 3b(xii) and Extended Data Fig. 7, have the following values (×100): (−58, −46, −34, −46) with statistical error 4. For comparison, the experimental values for the same set of stabilizers is (−52, −41, −39, −49). Our simulation results are in relatively good agreement with the measured data, suggesting that the model captures the effects of noise well. The observed discrepancies are expected to be due to inhomogeneity of the errors, which was not included in our error model. The simulations used an open source simulator qsim[46].

### Additional braiding data

In Fig. 3, we demonstrate that no fermion appears when distinguishable $\sigma$ are braided with each other. In Extended Data Fig. 8, we show the data for each step in that protocol, analogous to those shown for indistinguishable $\sigma$ in the main text. Moreover, we also present an alternative braiding scheme in Extended Data Fig. 9, which requires fewer (18) CZ gates. In this case, however, pair B is not brought back together, and neither of the $\sigma$ pairs are annihilated. Therefore, similar to in Fig. 2c, we measure the Pauli string corresponding to bringing a plaquette violation around pair A (grey path in Extended Data Fig. 9c), which in this case can be reduced to $\hat{Y}$ on the qubit where the two $\sigma$ overlap. We find $\langle \hat{P} \rangle = \langle \hat{Y} \rangle = -0.71 \pm 0.01$, indicating that braiding the $\sigma$ led to the creation of a fermion (Extended Data Fig. 9c). Note that we here only

search for fermions in one of the $\sigma$ pairs. As a control experiment, we repeat the experiment with distinguishable $\sigma$ pairs, as in the main text (Extended Data Fig. 9d). In this case, we find $\langle \hat{P} \rangle = +0.71 \pm 0.01$, thus demonstrating that no fermion was produced. Together, these observations constitute another demonstration of non-Abelian exchange statistics of the D3Vs.

## A summary of the theoretical framework

It was observed by Kitaev that fluxes of the e−m exchange symmetry are expected to host Majorana modes and therefore have the degeneracy of Ising anyons[9]. Bombin gave a particular stabilizer configuration realizing such a flux as a fixed lattice dislocation in a square grid, showed on general grounds that if such fluxes were well-separated and could be braided they would be projective Ising anyons, and noted that it may be possible to braid such fluxes by code deformation[10]. A general formalism for theories realized by braiding of symmetry fluxes was described in ref. 47. These constructions focus on the long-distance physics, and in practical terms[23] gives an account of 'microscopics'. An explicit mapping to a gauge theory shows how the anyons are localized to a single qubit, and is used to derive a simple, efficient and systematic procedure for creating, braiding and measuring the fusion outcomes of Ising anyons on general stabilizer graphs. The bridge between the microscopics and general arguments established by the gauge theory mapping allows us to fit several anyons on present-day devices, probe the full two-dimensional nature of their braiding by maintaining their separation, and demonstrate braid generators that restore all local observables. For details discussions of the protocol, see ref. 23.

## Data availability

The data that support the findings in this study are available at https://doi.org/10.5281/zenodo.7869220.

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

**Acknowledgements** Y.L. and E.A.K. acknowledge support by a New Frontier Grant from Cornell University's College of Arts and Sciences. E.A.K. acknowledges support by the National Science Foundation under grant no. OAC-2118310, the Ewha Frontier 10-10 Research Grant and the Simons Fellowship in Theoretical Physics award no. 920665. E.A.K. performed a part of this work at the Aspen Center for Physics, which is supported by the National Science Foundation grant no. PHY-160761.

**Author contributions** Y.D.L., K.K., E.-A.K. and I.A. developed the underlying theory. T.I.A., Y.D.L., K.K., E.-A.K., I.A. and P.R. developed the experiment. T.I.A. performed the experiment and analysed the data. I.K.D., A.B., S.H., A.M., X.M. and A.O. provided assistance with calibration. T.I.A., Y.D.L., K.K., E.-A.K., I.A. and P.R. wrote the manuscript. T.I.A., Y.D.L., K.K. and P.R. wrote the Methods. All authors contributed to revising the manuscript. All authors contributed to the experimental and theoretical infrastructure to enable the experiment.

**Competing interests** The authors declare no competing interests.

**Additional information**
**Correspondence and requests for materials** should be addressed to E.-A. Kim, I. Aleiner or P. Roushan.

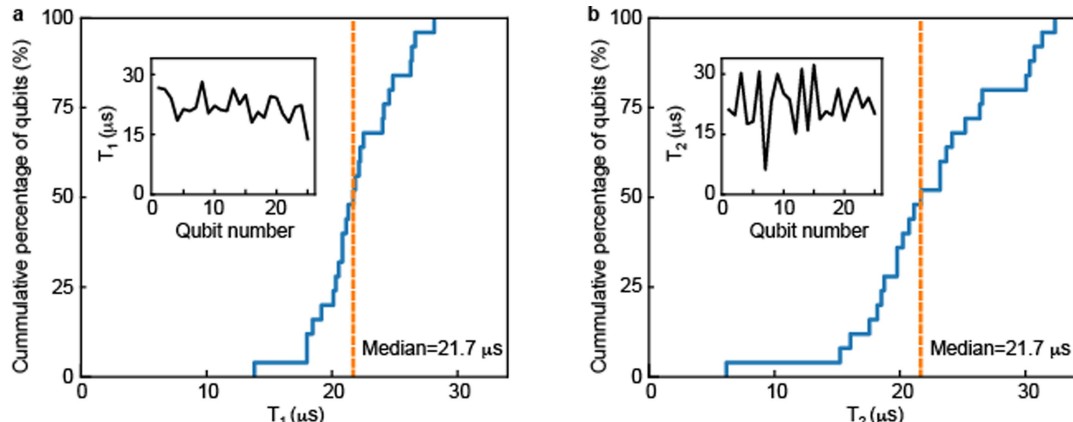

**Extended Data Fig. 1 | Qubit relaxation ($T_1$) and coherence ($T_2$) times. a,b**, Cummulative distributions of $T_1$ (**a**) and $T_2$ (**b**), where the latter is measured using a Hahn echo sequence. Dashed lines indicate the median values of 21.7 $\mu$s for both measures. Insets: $T_1$ and $T_2$ plotted against qubit number.

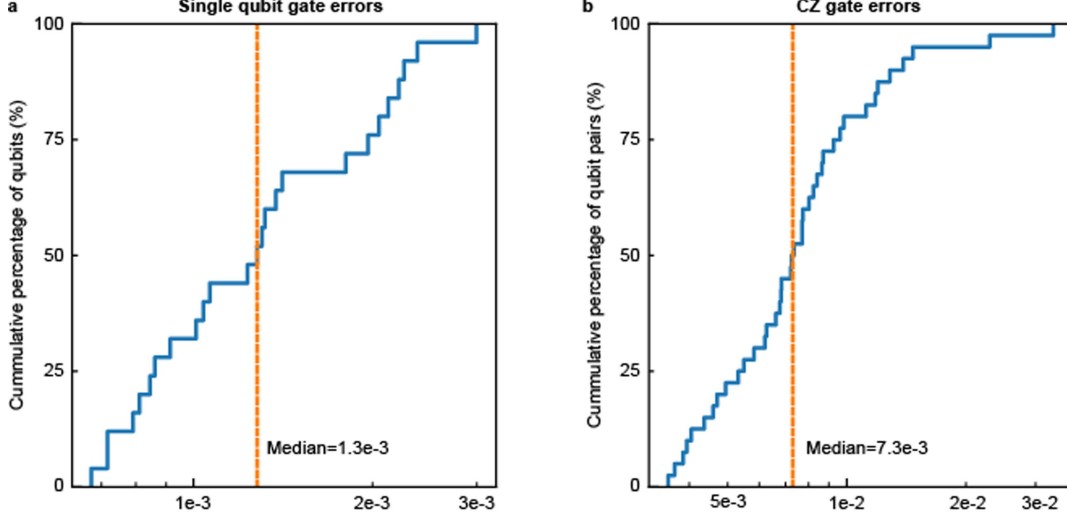

**Extended Data Fig. 2 | Gate errors. a**,**b**, Cummulative distributions of the Pauli error for single-qubit (**a**) and two-qubit CZ (**b**) gates. We find median error values of $1.3 \times 10^{-3}$ and $7.3 \times 10^{-3}$ for the single-qubit and CZ gates, respectively.

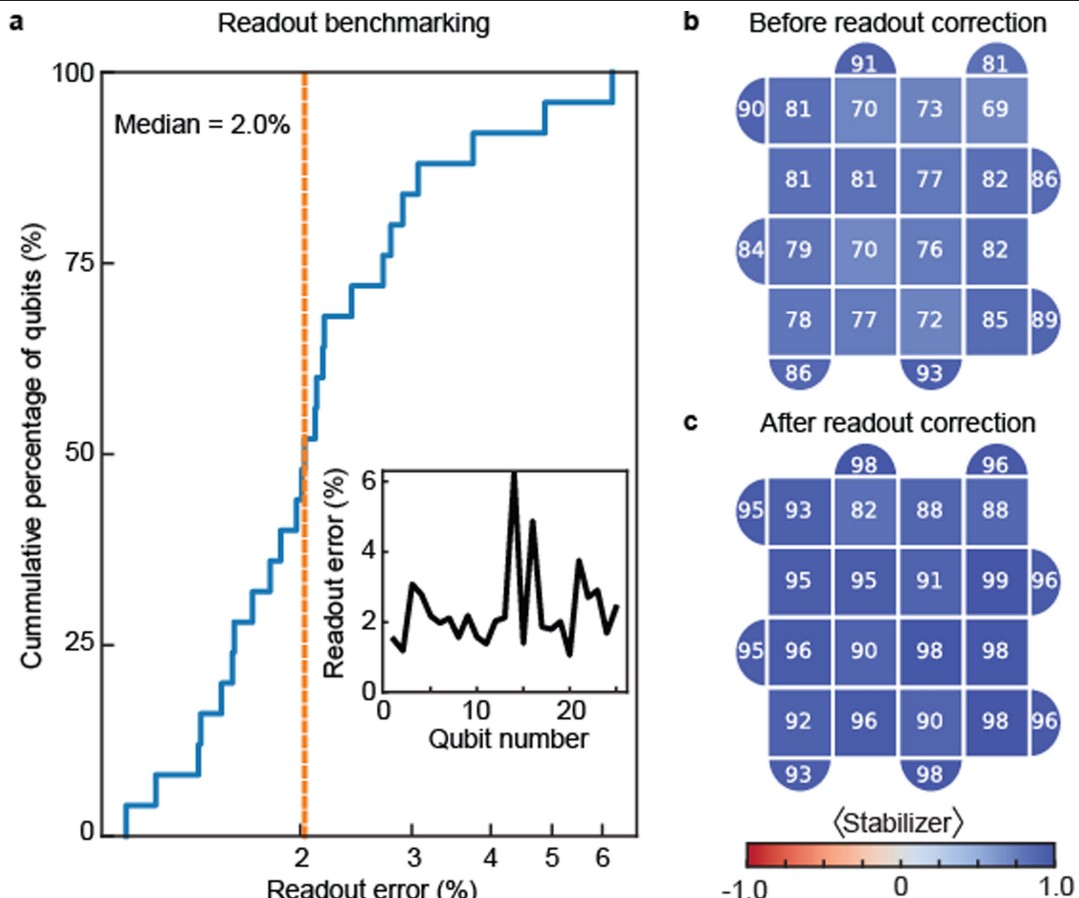

**Extended Data Fig. 3 | Readout benchmarking and correction. a**, Histogram of readout error, with a median value of 2.0% (dashed vertical line). Inset: readout error plotted against qubit number. **b,c**, Stabilizer values of the surface code ground state before (**b**) and after (**c**) readout correction.

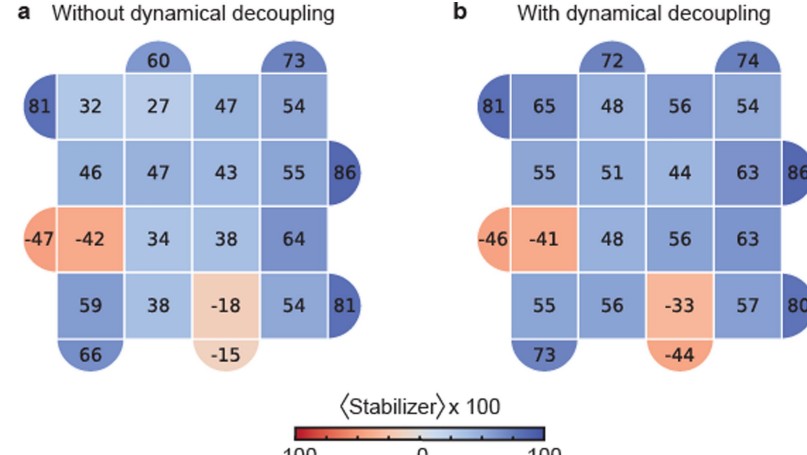

**Extended Data Fig. 4 | Dynamical decoupling. a,b**, Stabilizer values without (**a**) and with (**b**) dynamical decoupling, after D3V braiding. Dynamical decoupling improves the average absolute stabilizer value from 0.50 to 0.58.

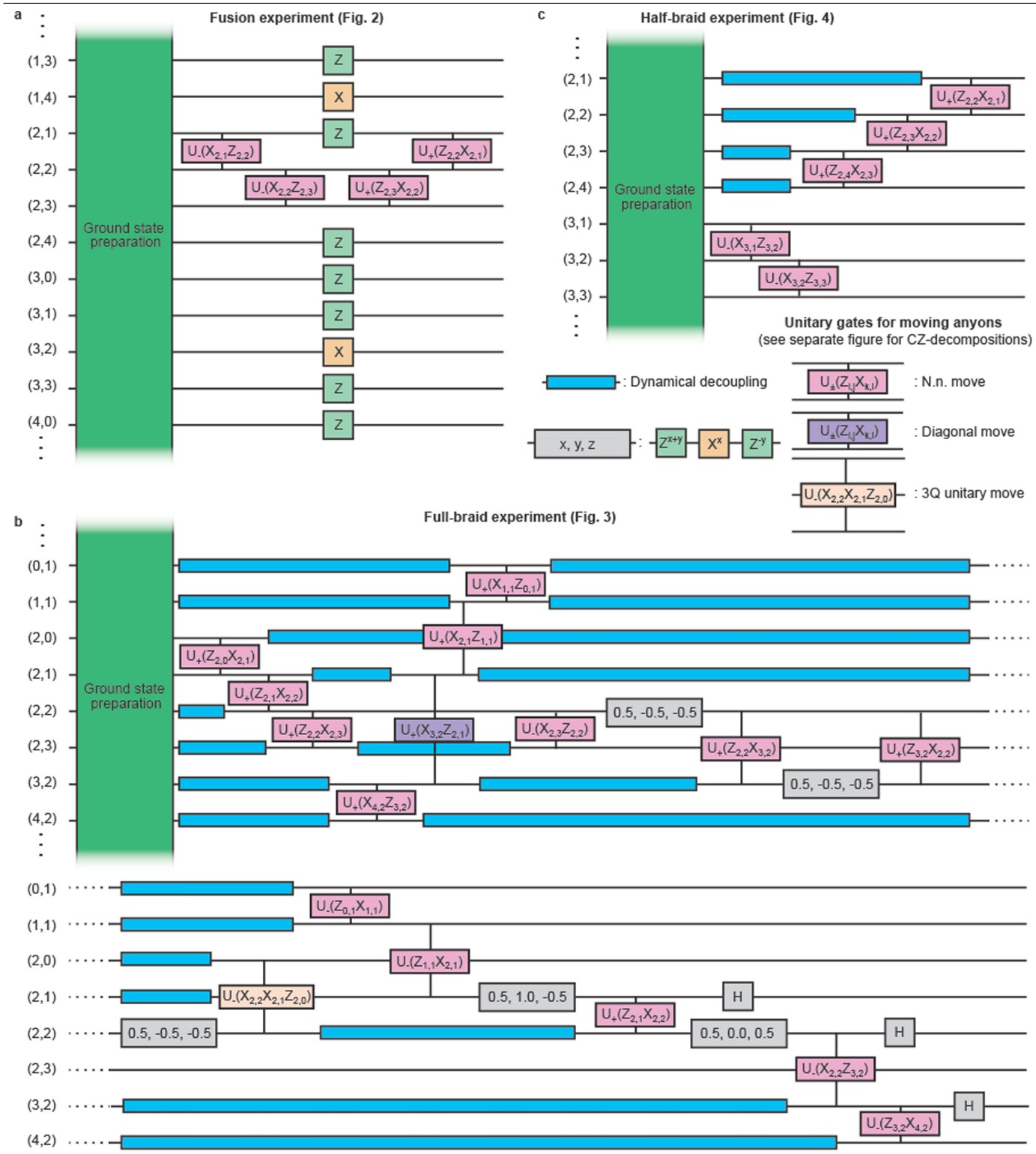

**Extended Data Fig. 5 | Circuit details.** Circuits used for the fusion experiment (**a**), the full-braid experiment (**b**), and the half-braid experiment (**c**), shown in Figs. 2–4, respectively, in the main text. Turqoise and gray boxes denote dynamical decoupling and phased $XZ$-gates, respectively. In the full-braid experiment (**b**), we include five single-qubit rotations to permute $\hat{X}$, $\hat{Y}$ and $\hat{Z}$ of the three stabilizers touching the moving D3V in steps V-VIII and IX-XI, as well as three Hadamard-gates to return all stabilizers to the original $\hat{Z}\hat{X}\hat{X}\hat{Z}$-form in XII. See Extended Data Fig. 6 for the circuit used for ground state preparation, as well as details on how the multi-qubit unitary gates used to move anyons are decomposed into CZ-gates.

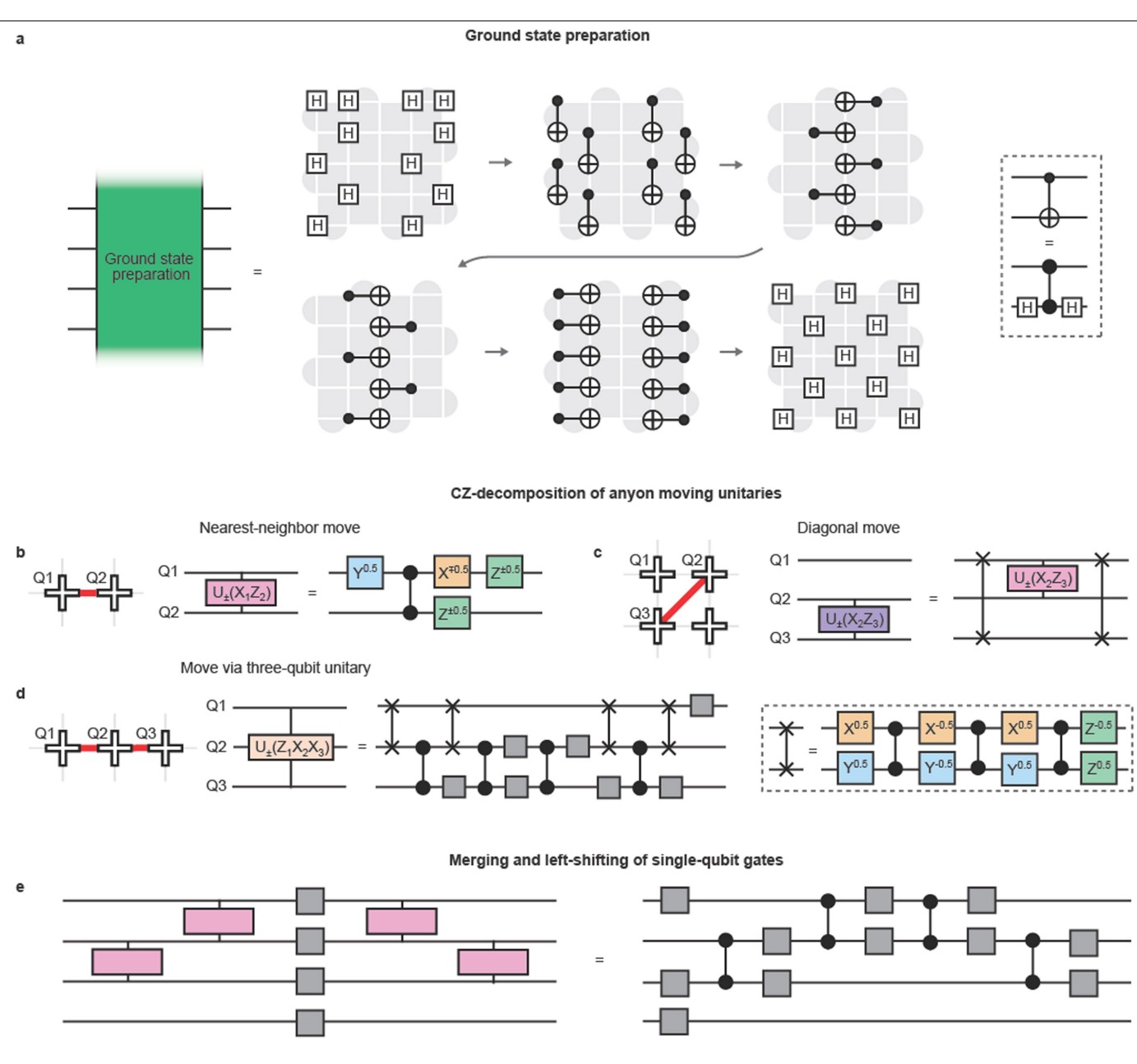

**Extended Data Fig. 6 | Ground state preparation and CZ-decompositions.**
**a**, Schematic showing the circuit used for preparation of the ground state of the surface code. The protocol is the same as that shown in ref. 43, except with the inclusion of Hadamard gates on alternating qubits in the final step, since we use symmetrized stabilizers on the form $\hat{Z}\hat{X}\hat{X}\hat{Z}$. **b**, The unitary needed to move a D3V between two neighboring vertices is realized in the experiment through the use of one CZ-gate and single-qubit rotations. **c**, When D3Vs are moved diagonally, we include two SWAP gates, requiring three CZ-gates each. **d**, Main: the three-qubit unitary used in step VIII in Fig. 3 is equivalent to a combination of single-qubit gates, 4 SWAP-gates and 4 CZ-gates. Right dashed box: decomposition of a SWAP-gate into CZ-gates. **e**, Adjacent single-qubit gates are merged and shifted left to the nearest CZ-gate.

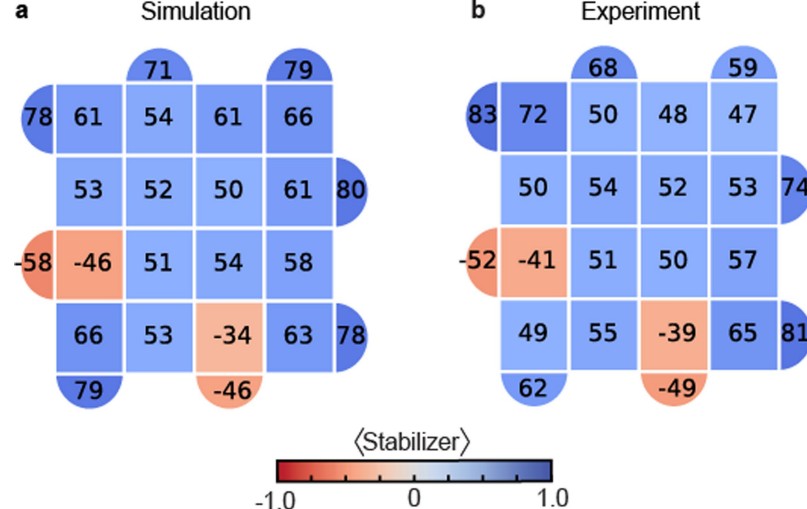

**Extended Data Fig. 7 | Simulation of braiding in the presence of noise. a**, Simulation results. **b**, Experimental data (same as in step XII in Fig. 3b). We observe relatively good agreement between the simulation and the experimental results, except some discrepancies that are attributed to inhomogeneity of the errors.

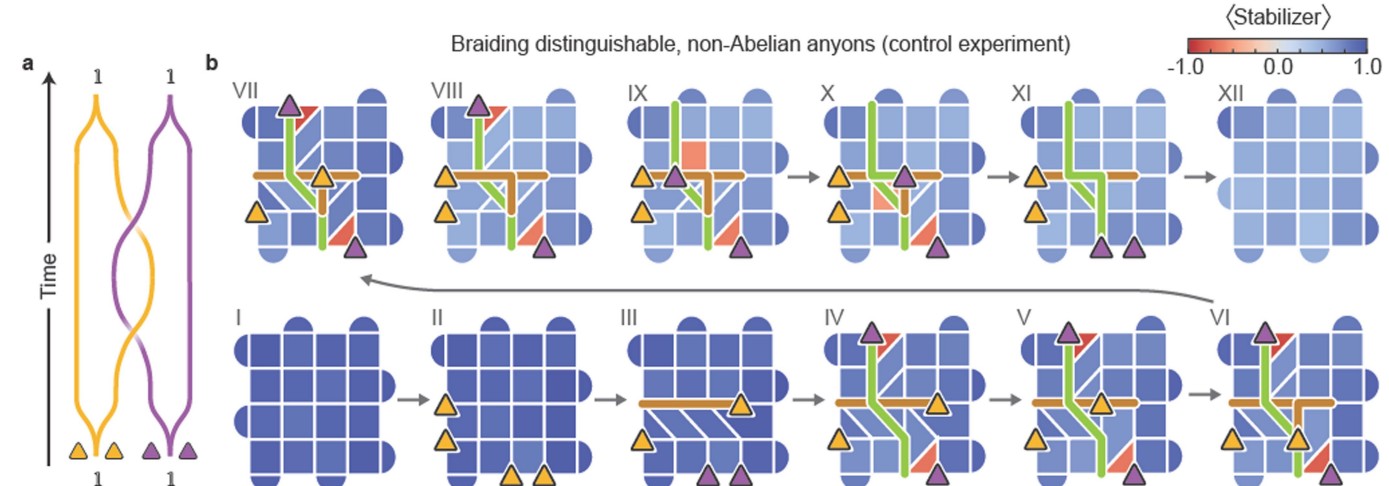

**Extended Data Fig. 8 | Braiding distinguishable D3Vs. a**, Braiding schematic of worldlines. **b**, Step-by-step depiction of stabilizers as the two $\sigma$ are braided, analogous to that in Fig. 3, but with distinguishable $\sigma$.

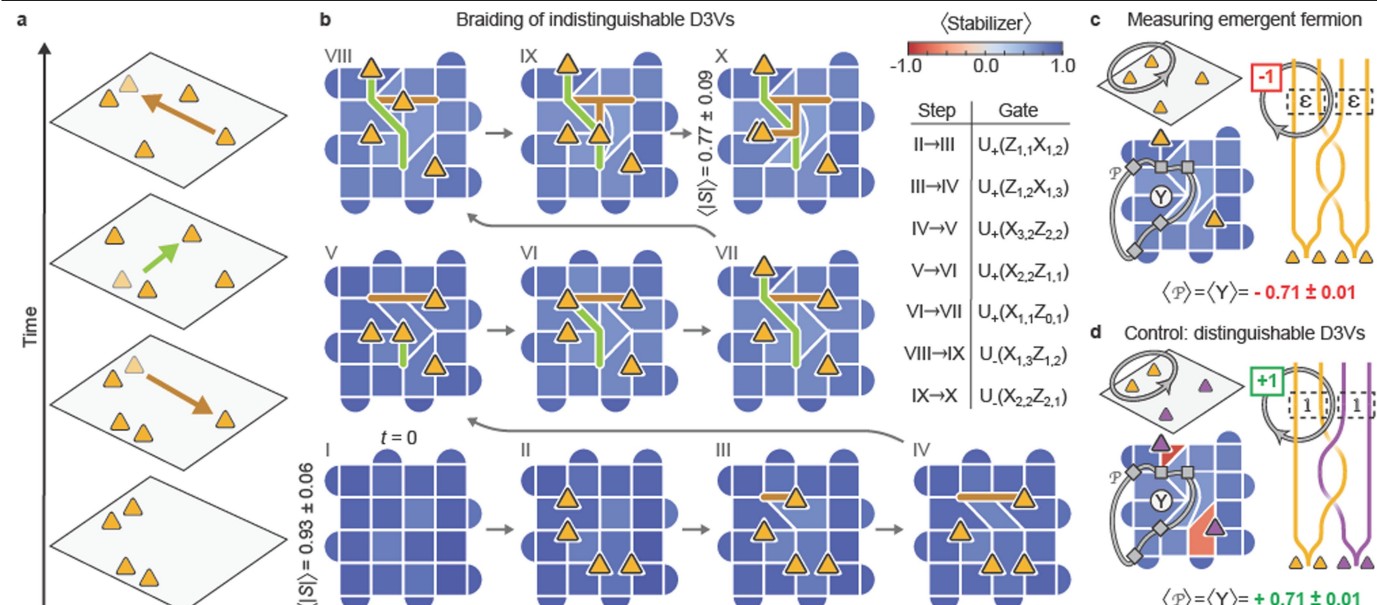

**Extended Data Fig. 9 | Alternative protocol for braiding σ. a**, Schematic displaying the braiding process of the two σ-pairs. **b**, Experimental demonstration of braiding, displaying the values of the stabilizers throughout the process. Two σ-pairs, A and B, are created from the vacuum 1, and one of the D3Vs in pair A is brought to the right side of the grid. Next, a σ from pair B is moved to the top, thus crossing the path of the first σ, before bringing the σ from pair A back again to complete the braid. The diagonal σ move performed in step VI is achieved by including two SWAP-gates, corresponding to 6 additional CZ-gates. The yellow triangles represent the locations of the σ, while the brown and green lines represent the paths of σ from pair A and B, respectively. The average absolute stabilizer value is 0.93 ± 0.06 and 0.77 ± 0.09 in the first and last step, respectively. **c**, After braiding the σ, we search for hidden fermions by measuring the Pauli string $\hat{\mathcal{P}}$ (left panels), which here is equivalent to $\hat{Y}$ on the qubit where the two σ overlap. The measurement yields $\langle\hat{\mathcal{P}}\rangle = \langle\hat{Y}\rangle = -0.71 \pm 0.01$, indicating creation of a fermion. Right: world-lines of braiding process, including non-local measurement based on plaquette violation loop. **d**, Same as **c**, but after braiding two distinguishable σ, achieved by applying the inverse two-qubit gates when moving the σ in pair B. The measurement yields $\langle\hat{Y}\rangle = +0.71 \pm 0.01$, indicating no fermion creation.