## [Peer Review File · Nature]

Manuscript Title: Non-Abelian braiding of graph vertices in a superconducting processor

Reviewer Comments & Author Rebuttals

Reviewer Reports on the Initial Version:

Referees' comments:

Referee #1 (Remarks to the Author):

This is an interesting report on a "software" implementation of Ising anyons. The possibility of such implementation is put forward by earlier theoretical considerations where the basic idea is to promote lattice defects of surface/toric code to quasi-particles. The experiment is conducted on a quantum processor that makes tens of qubits on a two-dimensional grid interact with one another via certain two- and one-qubit gates. All reported measurements concern certain observables on the system of qubits.

The authors advocate that they "experimentally verify the fusion rules of non-Abelian Ising anyons..." and furthermore "Our work represents a key step towards topological quantum computing," neither of which I agree.

Firstly, while the measurement itself is an interesting achievement, this is more like a numerical study when it comes to observation of emergent particles. Let BOX be their quantum processor; it could be any other box that can manipulate 25-qubit wavefunctions at user's disposal such as my laptop. They programmed BOX, and measured certain observables by interacting with BOX, where the outcomes are previously calculated modulo noise. The measurements are basically cross-checks to what pen-and-paper calculation shows. If I do a matrix multiplication on a piece of paper, and confirm it by running it on BOX, did I experimentally verify the multiplication rule of matrices? If I program equations of general relativity using BOX, and obtain some consistent measurement results inside the event horizon, should I claim that I have seen the interior of the blackhole? This kind of work is important if we are building a programmable device and is a nontrivial check of the device that it works as intended, but it says nothing further.

Secondly, the fact that one can implement surface code with lattice defects has no further implication for topological quantum computing than implementing the surface code alone. This experiment is about increasingly sophisticated ways to manipulate tens of qubits.

I would highly praise this experiment as a step towards a better quantum processor, but I cannot do so if the authors say this is observation of a previously elusive quasi-particle in a condensed matter system.

Referee #2 (Remarks to the Author):

Dear Sir/Madam,

The manuscript "Observation of non-Abelian exchange statistics on a superconducting processor" is an impressive piece of work and shows wonderful things that can be done on the best modern quantum computing devices.

It has been decades since prediction of braiding statistics and it is only recently that such braiding has been clearly observed in any experiment. This manuscript is for certain the first unambiguous observation of braiding of Majorana zero modes.

I think this is exciting and very important and should be published in Nature. If this cannot be published in Nature, nothing should be.

I do have some objection to this paper which should be easy to fix without a new experiment, just by rewriting text.

1) What is measured in this work is PROJECTIVE Ising anyons. This is NOT what people usually mean when they say non-Abelian anyons, although it is very similar. The coauthors who also wrote Ref 23 [arXiv:2210.09282] are aware of the distinction and are careful there to be precise. However in this manuscript the distinction is obscure. The distinction is crucial because it is not an anyon theory in the traditional sense. Most the experts in the field, and many who are not expert, do understand the difference. The experimental work is still fantastic, but should be clear about what is done.

2) It must also be made clear that while this may "highlight a new path of fault-tolerant quantum computation", it is not computationally universal yielding only Clifford gates.

3) The way credit is not given to other references is not acceptable. The fact that dislocations in the lattice (vertices of different degree) host Majorana zero modes was already stated in the groundbreaking work of Kitaev Ref 11. The fact that these can be braided was shown by Bombin Ref 9. These works are cited, but not clearly enough. They are cited in long lists of works that are not very relevant.

In fact this experiment can be summarized by saying that "We do exactly what Bombin proposed." The following from the paper

"..it was predicted by Lensky et al. that the vertices of degree 2 and 3 host non-Abelian anyons"

I think the community would find this statement offensive. Lensky's work may add to the literature, but this statement, and 98% of theory necessary for this paper, is already contained in Kitaev and Bombin papers. Also, point 1, it hosts PROJECTIVE non-Abelian anyons. Similar much of the discussion of how one should reconnect the lattice

"it was shown in Ref 23 that the D3Vs can in fact be moved"

This is in the Bombin paper too. Bombin cites some early papers that discuss lattice reconstruction in a different context, and similar things have been discussed many times since. The exact protocol used may be different than Bombin proposal, but this is details. The way it is stated gives credit to 23 and takes away deserved credit from Bombin.

In summary the paper should really say very clearly "THIS EXPERIMENT IS WHAT WAS PROPOSED BY BOMBIN/KITAEV". To do any less is very dishonest.

Although the experiment was proposed by Bombin, I must be clear however, that actually implementing the proposal is very difficult and very impressive. But credit should be given when it is due.

Despite my complaints I repeat that this is a great and very exciting experiment. Many people will be excited to see it!

Referee #3 (Remarks to the Author):

A: SUMMARY OF KEY RESULTS

The authors report on a sequence of beautiful experiments implemented on a superconducting quantum processor. First, the ground state of the surface code is prepared on 25 qubits with a shallow quantum circuit. Then, defects in the stabilizer graph are created, moved and exchanged via a sequence of unitary gates. A final measurement demonstrates that these manipulations correspond to braiding and fusing Ising anyons, which were theoretically expected to be associated with such defects. The braiding operation is also verified through quantum state tomography on an entangled state of three non-locally encoded qubits.

Overall, this series of protocols amounts to an experimental validation of the fusion and braiding rules of Ising anyons. However, I find it important to note that the protocols are carried out without quantum error correction (stabilizer measurements) and, thus, as far as I can tell, are not endowed with topological protection.

B. ORIGINALITY AND SIGNIFICANCE

In principle, the observation of non-Abelian statistics of Ising anyons would represent a huge breakthrough for quantum computing and, more broadly speaking, quantum physics. Thus, I have no doubt that these results could be of immediate scientific interest to many people in our discipline and surrounding ones.

However, I feel conflicted because the goal of observing non-Abelian statistics has been reached in this work in a rather narrow sense. Namely, the Authors have successfully checked that fusion rules

associated with defects in the surface code are obeyed. However, in my opinion the essence of quantum computing with non-Abelian anyons is that braiding outcomes are topologically protected: either "passively" (by the presence of an energy gap separating the computational states from all other states) or "actively" (by stabilizer measurements as in the surface code). I do not think that the results presented are topologically protected, in the following sense: I expect that if the Authors ran a similar braiding protocol on larger arrays of 6x6, 7x7, 8x8 qubits and so on, the final errors would not decrease exponentially. In fact, they may increase, because the braiding sequences would be longer and errors would have more time to accumulate.

There is no doubt to me that these experiments represent the state of the art and that, on the technical side, are an admirably difficult achievement in of themselves. It is natural however to compare this work to its more immediate predecessor, Ref. 43, published by Google Quantum AI in 2021 in Science. In that work the preparation of the surface code ground state was already demonstrated, together with a check of the fusion and braiding rules of its Abelian plaquette excitations. The present work adds the creation and braiding of non-Abelian defects. It is therefore a natural continuation of Ref. 43; one which, as far as I could tell, did not require significant additional theoretical or experimental breakthroughs compared to its predecessor. Viewed from this light, this work represents an exquisite cherry added to a cake that had been previously prepared.

While I find that the major claim of this work is correct, it is so in a way that is less consequential than one hopes. The desired topological protection of non-Abelian statistics has not been assessed. Topological quantum computing is still far away and would require running these protocols together with non-destructive stabilizer measurements. Hence, one may argue that the contemporary work from the same group, arXiv:2207.06431, on error correction in the surface code represents a more significant step towards topological quantum computing.

C: DATA AND METHODOLOGY

The Authors are succinct on some technical details regarding experimental protocols and the data processing. In particular:

1. A complete description of the quantum circuits performed is not provided; this extended data would be important to enable reproduction of the work on a comparable quantum processor.
2. I would ask the Authors to clarify the precise meaning of the reported expectation values of multi-qubit operators such as the stabilizers and the P , P' operators. Are these collected from sequences of destructive single-qubit measurements? That would be my guess given the qubit layout and the statement (page 2) that "the measurements are destructive and the protocol is restarted after each measurement".
3. What is the reasoning behind the correction applied to the stabilizer expectation values described in and around Fig. S3? Why is it required and justified?
4. Is it just a striking coincidence that the median T_1 and T_2 qubit times are both equal to 21.7 μs ? I would have thought that the qubits are relaxation-limited, but from the insets that does not quite seem to be the case. What is the dominant factor limiting T_2 times?
5. The expectation values of stabilizers seem to degrade way more in Fig. 3 than in Figs. 2 and 4. For instance, in step XII of Fig. 3 the stabilizers are (in absolute value) below 0.5, while they seem much

closer to 1 in step XI of Fig. 2b and step VII of Fig. 4c. Why this difference? Is it simply related to the length of the protocol?

D: APPROPRIATE USE OF STATISTICS AND TREATMENT OF UNCERTAINTIES

Two small points here:

1. I imagine that the expectation values of stabilizers reported in Figs. 2-4 originate from repeating the protocols multiple times. However, I was not able to find this information.
2. Please give a bit more details on the bootstrapping procedure used to estimate the uncertainty on the purity and overlap of the GHZ states in Fig. 4.

E: CONCLUSIONS: ROBUSTNESS, VALIDITY, RELIABILITY

The manuscript is, to the best of my comprehension, free of basic flaws. The experiments are well thought out and nicely conducted and described. The engineering is impressive.

The interpretation of the data is reliable. Some further insight on the origin of the errors limiting the final stabilizer measurements in the braiding protocol would be welcome.

F: SUGGESTED IMPROVEMENTS

Nothing to add to the previous list of questions. Expanding the manuscript to address the questions above would in my opinion benefit the readers.

G: REFERENCES

The list of references is appropriate.

H: CLARITY AND CONTEXT

The article reads very well, however, referring to my previous remarks on the topological protection, I find that in either introduction or conclusions the Authors should amplify the discussion on what are the steps that separate these results from topological quantum computing. At the moment, this is limited to only the last sentence of the paper: "With the potential inclusion of error correction..."

Similarly, in the introduction the authors write that "these platforms allow for multi-qubit (e.g. non-local) measurements". As remarked above, this sentence may lead the reader to misinterpret the results presented later, which if I understand correctly are all based on destructive measurements.

Finally, the discussion of the R matrix in the introduction could be made more precise. In particular, $R^2 = 1$ implies that $R = \pm 1$ only if R is a scalar. This fact is clarified only later in the paragraph. I'm sure the Authors know this better than me, but it just does not come across as clearly as it could.

SIGNED: Bernard van Heck

Author Rebuttals to Initial Comments:

Referee #1 (Remarks to the Author):

This is an interesting report on a "software" implementation of Ising anyons. The possibility of such implementation is put forward by earlier theoretical considerations where the basic idea is to promote lattice defects of surface/toric code to quasi-particles. The experiment is conducted on a quantum processor that makes tens of qubits on a two-dimensional grid interact with one another via certain two- and one-qubit gates. All reported measurements concern certain observables on the system of qubits.

The authors advocate that they "experimentally verify the fusion rules of non-Abelian Ising anyons..." and furthermore "Our work represents a key step towards topological quantum computing," neither of which I agree.

Firstly, while the measurement itself is an interesting achievement, this is more like a numerical study when it comes to observation of emergent particles. Let BOX be their quantum processor; it could be any other box that can manipulate 25-qubit wavefunctions at user's disposal such as my laptop. They programmed BOX, and measured certain observables by interacting with BOX, where the outcomes are previously calculated modulo noise. The measurements are basically cross-checks to what pen-and-paper calculation shows. If I do a matrix multiplication on a piece of paper, and confirm it by running it on BOX, did I experimentally verify the multiplication rule of matrices? If I program equations of general relativity using BOX, and obtain some consistent measurement results inside the event horizon, should I claim that I have seen the interior of the blackhole? This kind of work is important if we are building a programmable device and is a nontrivial check of the device that it works as intended, but it says nothing further.

We thank the Referee for taking the time to review our manuscript, and for providing us with the opportunity to clarify these important points.

We fully agree with the Referee that it is essential to discern physical demonstrations from mere matrix calculations. With regards to our case, there is a clear distinction from a computer simulation: while programming equations of general relativity would not imply that a black hole is present in the simulator, our main result (the non-Abelian exchange effects) does indeed occur in our processor. It fully relies on the quantum mechanical effects present in our system, and it physically occurs in the many-body wavefunction of microwave photons that we made. During our protocol, the anyons (manipulations of this many-body wavefunction) are actually moved around each other, in a spatially separated manner, and their braiding leads to a physical change in the local observables of our system. Because of this, our results are not merely another way to perform matrix calculations, but rather an actual, first observation of non-Abelian exchange statistics, realized as emergent properties of interacting microwave photons. We are therefore inclined to claim that we have experimentally verified the fusion rules of non-Abelian anyons, to which the other two referees seem to agree, describing our work as "*an experimental validation of the fusion and braiding rules of Ising anyons*" and "*the first unambiguous observation of braiding of Majorana zero modes.*"

We deeply thank the Referee for bringing our attention to the importance of the above clarification, and have now added the following sentence on page 1 to address it:

"We therefore emphasize that the 2D braiding processes are physically taking place on the device, leading to actual non-Abelian exchange effects of local anyons in the many-body wavefunction, rather than matrix operations that simply follow the same algebra."

The Referee is correct that the results in our manuscript could be readily theoretically predicted. However, we would like to argue that this is the case with many experimental advances and does not lower the impact of our results. On the contrary, we find it to be a major goal - and when successful, an exciting feat - in experimental physics to achieve the level of control required to realize good agreement between theoretical expectations and the physical outcome. Indeed, the significance of our work is in realizing this theoretically predicted physical phenomenon, which forms the backbone of topological quantum computation, on a noisy platform and showing that we have the necessary control and sufficiently low decoherence rates that the physics can be materialized. Another important aspect of our work is of course that it opens the door to exciting future directions that could go beyond simulatable physics. We emphasize this aspect of our work in the modified sentence on page 6:

[...] highlight a new path towards fault-tolerant implementation of Clifford gates, a key ingredient of universal quantum computation".

The Referee is also completely right that there are differences between our system and many of those that have traditionally been studied in the search for non-Abelian exchange statistics. While more conventional approaches typically involve Hamiltonian dynamics of quasi-particle excitations, we observe non-Abelian exchange statistics of the D3Vs by modifying the wavefunction through a sequence of unitary gates. (Note that the anyons are therefore not referred to as quasi-particles in our manuscript). Importantly, this distinction makes the observed exchange statistics neither less non-Abelian nor less physical. Nevertheless, we agree with the Referee that it is very important to describe the differences, and have therefore made several additions to address this, including:

"While efforts on conventional solid-state platforms typically involve Hamiltonian dynamics of quasi-particles, superconducting quantum processors allow for directly manipulating the many-body wavefunction of microwave photons via unitary gates." (abstract)

"Since the braiding of non-Abelian anyons in this platform is achieved through unitary gate control rather than adiabatic evolution of a Hamiltonian system, we note that the anyons are not quasi-particles in the sense of eigenstates that persist throughout a Hamiltonian evolution. Importantly, their movement is achieved through local operations along their paths, and they are kept spatially separated throughout the braiding." (page 1)

"In other, more conventional candidate platforms for non-Abelian exchange statistics, which involve Hamiltonian dynamics of quasi-particle excitations, topological protection naturally arises from an emergent gap that separates the computational states from other states. In order to leverage the non-Abelian anyons in our system for topologically protected quantum computing, the stabilizers must be measured throughout the braiding protocol." (page 6)

We thank the Referee for motivating these very important additions!

Secondly, the fact that one can implement surface code with lattice defects has no further implication for topological quantum computing than implementing the surface code alone. This experiment is about increasingly sophisticated ways to manipulate tens of qubits.

We respectfully disagree with the Referee's statement, since implementing the surface code with lattice defects involves numerous drastic distinctions from the regular surface code, that can not only prove advantageous in the development of topological quantum computing, but are also of interest from a purely scientific point of view. The demonstrated control of the anyons, including their creation, braiding and fusion, enables new ways of storing and processing quantum information, including for instance:

-Two-qubit gates performed through braiding of D3Vs instead of lattice surgery

- Dynamically adjustable code distance by moving the anyons
- Controllable density of encoded qubits by creating and fusing anyons

We therefore think that the great attention that the scientific community has given to theoretical proposals of lattice defects is justified, and that our experimental demonstration will be of wide interest to the field. In general, it is of great importance to study alternative approaches to achieving scientific and technological goals, so long as they carry justifiable promise. We confidently believe that this is the case with the approach studied in our work.

In an effort to be as clear as possible with regards to the advances that were made in our work, we have edited the last sentence of the abstract to:

"Our work realizes key components of topological quantum computing."

I would highly praise this experiment as a step towards a better quantum processor, but I cannot do so if the authors say this is observation of a previously elusive quasi-particle in a condensed matter system.

We thank the Referee for complimenting the aspect of our work that pertains to the development of quantum processors. Indeed, this is an important part of our work, which is discussed in detail in the supplement. Since the realization of non-Abelian exchange statistics is the most central aspect of this particular study, we have chosen to predominantly focus on that in the main text.

There are certainly differences between our system and traditional condensed matter systems studied in the search for non-Abelian exchange statistics, and in agreement with the Referee's comment, we have ensured not to refer to the D3Vs as quasi-particles. However, as discussed above, these differences do not imply that the exchange statistics studied in our work are any less non-Abelian or physical. Physically distinct from calculations on a piece of paper or computer simulations, the exchange effects actually occur in our device, when we manipulate the quantum many-body wavefunction of microwave photons. We have done our best to discuss the differences between our system and more conventional ones in the revised manuscript, and thank the Referee for motivating this addition!

Referee #2 (Remarks to the Author):

Dear Sir/Madam,

The manuscript "Observation of non-Abelian exchange statistics on a superconducting processor" is an impressive piece of work and shows wonderful things that can be done on the best modern quantum computing devices.

It has been decades since prediction of braiding statistics and it is only recently that such braiding has been clearly observed in any experiment. This manuscript is for certain the first unambiguous observation of braiding of Majorana zero modes. I think this is exciting and very important and should be published in Nature. If this cannot be published in Nature, nothing should be.

We thank the Referee for emphasizing the importance of our work and for recommending that it should be published in Nature. We also want to thank the Referee for the very helpful comments, which certainly helped us improve the manuscript.

I do have some objection to this paper which should be easy to fix without a new experiment, just by rewriting text.

1) What is measured in this work is PROJECTIVE Ising anyons. This is NOT what people usually mean when they say non-Abelian anyons, although it is very similar. The coauthors who also wrote Ref 23 [arXiv:2210.09282] are aware of the distinction and are careful there to be precise. However in this manuscript the distinction is obscure. The distinction is crucial because it is not an anyon theory in the traditional sense. Most the experts in the field, and many who are not expert, do understand the difference. The experimental work is still fantastic, but should be clear about what is done.

We agree with the Referee that this is an important point, which we have now addressed through the following sentences on page 2 in the main text:

"[...] can host projective non-Abelian Ising anyons. For brevity, we refer to these as "non-Abelian anyons" or simply "anyons" from here on."

We have also ensured to write "projective non-Abelian Ising anyons" in the abstract.

2) It must also be made clear that while this may "highlight a new path of fault-tolerant quantum computation", it is not computationally universal yielding only Clifford gates.

We thank the Referee for raising this great point, and have now changed the sentence in question to:

"[...] highlight a new path towards fault-tolerant implementation of Clifford gates, a key ingredient of universal quantum computation".

3) The way credit is not given to other references is not acceptable. The fact that dislocations in the lattice (vertices of different degree) host Majorana zero modes was already stated in the groundbreaking work of Kitaev Ref 11. The fact that these can be braided was shown by Bombin Ref 9. These works are cited, but not clearly enough. They are cited in long lists of works that are not very relevant.

We fully agree with the Referee that the works of Kitaev and Bombin deserve clear credit, and have now made several edits to emphasize their contributions, listed below as responses to the relevant points. We have also added the following paragraph as a new section VII in the Supplementary Material, in order to give a clear account of the contributions from Bombin, Kitaev and others:

"It was observed by Kitaev that fluxes of the e-m exchange symmetry are expected to host Majorana modes and therefore have the degeneracy of Ising anyons [Kitaev2006]. Bombin gave a particular stabilizer configuration realizing such a flux as a fixed lattice dislocation in a square grid, showed on general grounds that if such fluxes were well-separated and could be braided they would be projective Ising anyons, and noted that it may be possible to braid such fluxes by code deformation [Bombin2010]. A general formalism for theories realized by braiding of symmetry fluxes was described in [Barkeshli2019]. These constructions focus on the long-distance physics, and in practical terms [Lensky] gives an account of "microscopics". An explicit mapping to a gauge theory shows how the anyons are localized to a single qubit, and is used to derive a simple, efficient, and systematic procedure for creating, braiding, and measuring the fusion outcomes of Ising anyons on general stabilizer graphs. The bridge between the microscopics and general arguments established by the gauge theory mapping allows us to fit several anyons on present-day devices, probe the full 2-dimensional nature of their braiding by maintaining their separation, and demonstrate braid generators which restore all local observables. For details discussions of the protocol see [Lensky]."

In fact this experiment can be summarized by saying that "We do exactly what Bombin proposed." The following from the paper

"..it was predicted by Lensky et al. that the vertices of degree 2 and 3 host non-Abelian anyons".

I think the community would find this statement offensive. Lensky's work may add to the literature, but this statement, and 98% of theory necessary for this paper, is already contained in Kitaev and Bombin papers. Also, point 1, it hosts PROJECTIVE non-Abelian anyons.

This sentence has been changed to the following: *"To realize non-Abelian statistics, one needs to go beyond such plaquette violations; it has been proposed that dislocations in the stabilizer graph - analogous to lattice defects in crystalline solids - can host projective non-Abelian Ising anyons [Kitaev, Bombin]. For brevity, we refer to these as "non-Abelian anyons" or simply "anyons" from here on.*

In the graph framework introduced above, it has been shown that such dislocations are characterized as vertices of degree 2 and 3 [Lensky]."

Similar much of the discussion of how one should reconnect the lattice "it was shown in Ref 23 that the D3Vs can in fact be moved". This is in the Bombin paper too. Bombin cites some early papers that discuss lattice reconstruction in a different context, and similar things have been discussed many times since. The exact protocol used may be different than Bombin proposal, but this is details. The way it is stated gives credit to 23 and takes away deserved credit from Bombin.

We have edited this part to: *"In order to be braided and fused by unitary operations, the D3Vs must be moved. While the structure of the stabilizer graph is usually considered to be static, it was predicted by Bombin [Bombin] that the dislocations in the surface code would exhibit projective non-Abelian Ising statistics if braided. Here, we will employ a specific protocol recently proposed by Lensky et al. [Lensky] for deforming the stabilizer graph (and thus moving the anyons) using local two-qubit Clifford gates."*

In summary the paper should really say very clearly "THIS EXPERIMENT IS WHAT WAS PROPOSED BY BOMBIN/KITAEV". To do any less is very dishonest.

On page 2, we have now added the sentence: *"Following these insights from Kitaev and Bombin, we now turn to our experimental study of the proposed anyons, using the protocol described in Ref. [Lensky]."*

Although the experiment was proposed by Bombin, I must be clear however, that actually implementing the proposal is very difficult and very impressive. But credit should be given when it is due.

Despite my complaints I repeat that this is a great and very exciting experiment. Many people will be excited to see it!

We again thank the Referee for the very helpful comments and for the kind compliments regarding the quality and impact of our work!

Referee #3 (Remarks to the Author):

A: SUMMARY OF KEY RESULTS

The authors report on a sequence of beautiful experiments implemented on a superconducting quantum processor. First, the ground state of the surface code is prepared on 25 qubits with a shallow quantum circuit. Then, defects in the stabilizer graph are created, moved and exchanged via a sequence of unitary gates. A final measurement demonstrates that these manipulations correspond to braiding and fusing Ising anyons, which were theoretically expected to be associated with such defects. The braiding operation is also verified through quantum state tomography on an entangled state of three non-locally encoded qubits.

Overall, this series of protocols amounts to an experimental validation of the fusion and braiding rules of Ising anyons. However, I find it important to note that the protocols are carried out without quantum error correction (stabilizer measurements) and, thus, as far as I can tell, are not endowed with topological protection.

We thank the Referee for complimenting our work, and for their detailed comments, which helped us make important improvements to the manuscript. Our response and corresponding edits can be found below, where we also address the point about error correction.

B. ORIGINALITY AND SIGNIFICANCE

In principle, the observation of non-Abelian statistics of Ising anyons would represent a huge breakthrough for quantum computing and, more broadly speaking, quantum physics. Thus, I have no doubt that these results could be of immediate scientific interest to many people in our discipline and surrounding ones.

However, I feel conflicted because the goal of observing non-Abelian statistics has been reached in this work in a rather narrow sense. Namely, the Authors have successfully checked that fusion rules associated with defects in the surface code are obeyed. However, in my opinion the essence of quantum computing with non-Abelian anyons is that braiding outcomes are topologically protected: either "passively" (by the presence of an energy gap separating the computational states from all other states) or "actively" (by stabilizer measurements as in the surface code). I do not think that the results presented are topologically protected, in the following sense: I expect that if the Authors ran a similar braiding protocol on larger arrays of 6x6, 7x7, 8x8 qubits and so on, the final errors would not decrease exponentially. In fact, they may increase, because the braiding sequences would be longer and errors would have more time to accumulate.

The Referee is correct that our work does not include error correction, and we agree that this is an exciting future step towards topological quantum computing (TQC). However, we would argue that the observation of non-Abelian exchange statistics is still a very impactful discovery for two main reasons. First, it involves very novel physics that stands in contrast to the behavior of all particles observed to date. In other words, even in a hypothetical case where non-Abelian anyons did not carry promise for TQC, the braiding behavior of non-Abelian anyons is a very interesting phenomenon that deserves wide attention in itself. Second, as with most great feats in science, the realization of TQC will certainly consist of multiple sequential leaps. Seeing as the observation of non-Abelian exchange statistics is such an essential step in this sequence and has been a major scientific goal for decades, we are confident that our work will be highly impactful and spark a lot of interest in the community.

In the revised main text, we now discuss the point about topological protection, as well as how it arises in different systems, in the following sentences on page 6:

"In other, more conventional candidate platforms for non-Abelian exchange statistics, which involve Hamiltonian dynamics of quasi-particle excitations, topological protection naturally arises from an emergent gap that separates the computational states from other states. In order to leverage the non-Abelian anyons in our system for topologically protected quantum computing, the stabilizers must be measured throughout the braiding protocol. With the potential inclusion of this error correction procedure, which involves overheads including readout of 5-qubit stabilizers, our observations highlight a new path towards fault-tolerant implementation of Clifford gates, a key ingredient of universal quantum computation"

Moreover, in order to be as clear as possible about the advances made in our work, we have also edited the final sentence of the abstract to:

"Our work realizes key components of topological quantum computing."

There is no doubt to me that these experiments represent the state of the art and that, on the technical side, are an admirably difficult achievement in of themselves. It is natural however to compare this work to its more immediate predecessor, Ref. 43, published by Google Quantum AI in 2021 in Science. In that work the preparation of the surface code ground state was already demonstrated, together with a check of the fusion and braiding rules of its Abelian plaquette excitations. The present work adds the creation and braiding of non-Abelian defects. It is therefore a natural continuation of Ref. 43; one which, as far as I could tell, did not require significant additional theoretical or experimental breakthroughs compared to its predecessor. Viewed from this light, this work represents an exquisite cherry added to a cake that had been previously prepared.

The Referee is absolutely right that the work in Science in 2021 also concerns the surface code and that the initial ground state is prepared in the same way. However, there are several strong reasons why the demonstration of non-Abelian exchange statistics is a very different discovery:

- Most importantly, the braiding of non-Abelian anyons is fundamentally different from that of Abelian ones, both from the perspective of the underlying physics and potential applications. The fact that Abelian anyons were observed previously certainly does not diminish the impact of observing their non-Abelian counterparts.
- In terms of implementation, the only commonality between the two works is the preparation of the ground state. In the present work, the crucial breakthrough is not the ground state preparation, but the procedure used for moving the D3Vs; there is no similar operation in the previous work.
- The idea of dynamically reshaping the stabilizer graph is fundamentally different from moving excitations on a static graph.

For these reasons, we are confident that our work provides very impactful and novel insights.

While I find that the major claim of this work is correct, it is so in a way that is less consequential than one hopes. The desired topological protection of non-Abelian statistics has not been assessed. Topological quantum computing is still far away and would require running these protocols together with non-destructive stabilizer measurements. Hence, one may argue that the contemporary work from the same group, arXiv:2207.06431, on error correction in the surface code represents a more significant step towards topological quantum computing.

As discussed above, we agree that the addition of non-destructive stabilizer measurements will be an exciting future direction, but would argue that the observation of non-Abelian exchange statistics is in itself a major scientific discovery. Efforts on the conventional surface code have certainly been around for much longer and are more established. The Referee is therefore completely right that our team's work on this direction is further developed. This is of course only natural, since the non-Abelian anyon approach is both newer and less studied. We do not believe that it diminishes the impact of our work; instead, our observations open the door towards an exciting alternative direction for TQC, and are of course very interesting from a more general physics perspective as well.

C: DATA AND METHODOLOGY

The Authors are succinct on some technical details regarding experimental protocols and the data processing. In particular:

1. A complete description of the quantum circuits performed is not provided; this extended data would be important to enable reproduction of the work on a comparable quantum processor.

We thank the Referee for this very valuable suggestion. In the newly added Supplementary Fig. S5, we now show the circuits used for all the experiments presented in the main text. Moreover, the diagrams in

Supplementary Fig. S6 provide the information needed to produce more general braiding circuits. For the Referee's convenience, we have reproduced both of these figures below:

Fig. S5: Circuit details. Circuits used for the fusion experiment (a), the full-braid experiment (b), and the half-braid experiment (c), shown in Figs. 2-4, respectively, in the main text. Turquoise and gray boxes denote dynamical decoupling and phased XZ-gates, respectively. In the full-braid experiment (b), we include five single-qubit rotations to permute X, Y and Z of the three stabilizers touching the moving D3V in steps V-VIII and IX-XI, as well as three Hadamard-gates to return all stabilizers to the original ZXXZ-form in XII. See Fig. S6 for the circuit used for ground state preparation, as well as details on how the multi-qubit unitary gates used to move anyons are decomposed into CZ-gates.

Fig. S6: Ground state preparation and CZ-decompositions. *a*, Schematic showing the circuit used for preparation of the ground state of the surface code. The protocol is the same as that shown in [Satzinger et al., 2021], except with the inclusion of Hadamard gates on alternating qubits in the final step, since we use symmetrized stabilizers on the form $ZXXZ$. *b*, The unitary needed to move a $D3V$ between two neighboring vertices is realized in the experiment through the use of one CZ-gate and single-qubit rotations. *c*, When $D3Vs$ are moved diagonally, we include two SWAP-gates, requiring three CZ-gates each. *d*, Main: The three-qubit unitary used in step VIII in Fig. 3 is equivalent to a combination of single-qubit gates, 4 SWAP-gates and 4 CZ-gates. Right dashed box: decomposition of a SWAP-gate into CZ-gates. *e*, Adjacent single-qubit gates are merged and shifted left to the nearest CZ-gate.

2. I would ask the Authors to clarify the precise meaning of the reported expectation values of multi-qubit operators such as the stabilizers and the P, P' operators. Are these collected from sequences of destructive single-qubit measurements? That would be my guess given the qubit layout and the statement (page 2) that "the measurements are destructive and the protocol is restarted after each measurement".

We thank the Referee for pointing out this potential source of confusion. As the Referee suggested, these quantities are determined by destructively measuring the involved qubits in their relevant bases simultaneously. For instance, $\langle X_1 Z_2 \rangle$ is determined by repeatedly measuring qubit 1 (2) in the X(Z) basis and averaging their product. We have now addressed this point by adding the following sentence on page 2:

[...] (determined by simultaneously measuring the involved qubits in their respective bases, $n=10,000$; note that the measurements are destructive and the protocol is restarted after each measurement).

3. What is the reasoning behind the correction applied to the stabilizer expectation values described in and around Fig. S3? Why is it required and justified?

Since the readout of the qubit state is imperfect, the raw data gives a somewhat incorrect representation of the actual quantum state of the system. We write the probability of readout error of state 0(1) on qubit i as $p_{0(1),i}$, and the readout fidelity is thus given by $1-(p_{0,i}+p_{1,i})/2$. In order to correct for any asymmetry between readout of the 0 and 1 states, we perform symmetrized measurements in which π -pulses are applied to the qubits before the readout in half of the measurements and the recorded qubit values are inverted. The measured value of a stabilizer with actual value $\langle S \rangle = \langle \prod \alpha_i \rangle$ is then:

$$\langle S \rangle_{\text{meas}} = \langle \prod (1-p_{0,i}-p_{1,i}) \alpha_i \rangle = \prod (1-p_{0,i}-p_{1,i}) \langle S \rangle,$$

where we made use of the fact that each qubit is measured equally often in the 0- and 1-states in the symmetrized measurement scheme. Note the absence of the factor $1/2$ compared to the expression for the readout fidelity, since perfectly incorrect readout ($p_0=p_1=1$) would give a readout fidelity of 0, but a measured value of $-\alpha_i$. In order to correct for the discrepancy between the measured stabilizer value and the actual stabilizer value, we measure $\langle Z_1 \dots Z_n \rangle$ of the state $|00\dots 00\rangle$ with the same qubits (using again symmetrized measurements) to find:

$$\langle Z_1 \dots Z_n \rangle_{\text{meas}} = \prod (1-p_{0,i}-p_{1,i}).$$

The readout-corrected $\langle S \rangle$ is then found from:

$$\langle S \rangle = \langle S \rangle_{\text{meas}} / \langle Z_1 \dots Z_n \rangle_{\text{meas}}$$

We thank the Referee for pointing out that this was not clear, and have now added the above explanation to section II in the Supplementary Material.

4. Is it just a striking coincidence that the median T1 and T2 qubit times are both equal to 21.7 μs ? I would have thought that the qubits are relaxation-limited, but from the insets that does not quite seem to be the case. What is the dominant factor limiting T2 times?

It is correct that the median T1 and T2 times are both equal to 21.7 μs , which is simply a coincidence. The difference between the measured decoherence rate $1/T_2$ and the relaxation-limited value $1/(2T_1)$ is due to remnant noise that is not decoupled in the Hahn echo experiment (in Ramsey experiments, the dephasing time T_2^* is typically on the order of 5 μs). In other words, T2 is approximately equally affected by relaxation and remnant lower-frequency noise. We thank the Referee for this very good question and have addressed it by adding the following sentence to section I in the Supplementary Material:

"We note that the equality of T1 and T2 is a coincidence and that the discrepancy between the measured decoherence rate $1/T_2$ and the relaxation-limited rate $1/(2T_1)$ is due to remnant noise not decoupled in the Hahn echo experiment."

5. The expectation values of stabilizers seem to degrade way more in Fig. 3 than in Figs. 2 and 4. For instance, in step XII of Fig. 3 the stabilizers are (in absolute value) below 0.5, while they seem much closer to 1 in step XI of Fig. 2b and step VII of Fig. 4c. Why this difference? Is it simply related to the length of the protocol?

The Referee is completely right that the difference in degradation between Figs. 2 and 4, and Fig. 3 is due to the different lengths of the protocols. While Figs. 2 and 4 only involve, respectively, 4 and 5 CZ-gates for moving the D3Vs, Fig. 3 contains 36. This is partially due to diagonal move in step IV (10 CZ-gates) and the three-qubit unitary in step VIII (25 CZ-gates). Besides the description of the circuit depth in the caption of Fig. 3, the difference in the depths of the circuits can now be clearly seen in the newly added Fig. S5.

To further illustrate the effects of noise on the output of the braiding experiment, we have performed an approximate simulation of the density matrix evolution of the system under a model of noise parametrized by T1, T2 and gate fidelities measured in independent experiments. This model captures most of the effects of noise observed in the experiment. We have added a new section V to describe this simulation in the Supplementary Material.

D: APPROPRIATE USE OF STATISTICS AND TREATMENT OF UNCERTAINTIES

Two small points here:

1. I imagine that the expectation values of stabilizers reported in Figs. 2-4 originate from repeating the protocols multiple times. However, I was not able to find this information.

The expectation values are measured by repeating the protocol 10,000 times. We thank the Referee for pointing out that this information was missing, and have now added it in the following sentence on page 2:

“(determined by simultaneously measuring the involved qubits in their respective bases, $n=10,000$; [..])”

2. Please give a bit more details on the bootstrapping procedure used to estimate the uncertainty on the purity and overlap of the GHZ states in Fig. 4.

The bootstrapping procedure was performed by resampling the data 10,000 times from the original data set. We thank the Referee for pointing out that this was unclear, and have now added the following clarification on page 6:

“[...] uncertainties estimated from bootstrapping method; resampled 10,000 times from the original data set”

E: CONCLUSIONS: ROBUSTNESS, VALIDITY, RELIABILITY

The manuscript is, to the best of my comprehension, free of basic flaws. The experiments are well thought out and nicely conducted and described. The engineering is impressive.

The interpretation of the data is reliable. Some further insight on the origin of the errors limiting the final stabilizer measurements in the braiding protocol would be welcome.

We thank the Referee for these kind compliments to our work, and as mentioned above, we have now also performed simulations to study the effects of noise (section V in the Supplementary Material).

F: SUGGESTED IMPROVEMENTS

Nothing to add to the previous list of questions. Expanding the manuscript to address the questions above would in my opinion benefit the readers.

As described above, we have made numerous edits to address all of the Referee's concerns to the best of our ability, and thank the Referee for the very valuable suggestions!

G: REFERENCES

The list of references is appropriate.

H: CLARITY AND CONTEXT

The article reads very well, however, referring to my previous remarks on the topological protection, I find that in either introduction or conclusions the Authors should amplify the discussion on what are the steps that separate these results from topological quantum computing. At the moment, this is limited to only the last sentence of the paper: "With the potential inclusion of error correction..."

As discussed above, we have now made edits to address this point. We hope that it is now more clear, and thank the Referee for the valuable suggestion!

Similarly, in the introduction the authors write that "these platforms allow for multi-qubit (e.g. non-local) measurements". As remarked above, this sentence may lead the reader to misinterpret the results presented later, which if I understand correctly are all based on destructive measurements.

We completely agree with the Referee that this was a potential source of confusion, and have now edited the sentence to:

"Moreover, these platforms allow for probing arbitrary Pauli strings through destructive multi-qubit (i.e. non-local) measurements."

Finally, the discussion of the R matrix in the introduction could be made more precise. In particular, $R^2 = 1$ implies that $R = \pm 1$ only if R is a scalar. This fact is clarified only later in the paragraph. I'm sure the Authors know this better than me, but it just does not come across as clearly as it could.

We thank the Referee for raising this very good point! The relevant sentence on page 1 has now been changed to:

"[...], it is thus required that $R^2=1$ (a scalar), [...]"

SIGNED: Bernard van Heck

Reviewer Reports on the First Revision:

Referees' comments:

Referee #1 (Remarks to the Author):

As long as the paper contains any allusion that they first experimentally observe nonabelian fusion/braiding rules or the conclusion that they realize key components of topological quantum computing, I do not recommend this manuscript for publication in any journal, not just Nature.

The authors have toned down their claim about nonabelian fusion/braiding, and certainly the new title is much more honest. Still, I do not think it is fully satisfactory. The paper starts by citing the long history of nonabelian anyons in 2+1d, called some quasiparticles anyons. (In the rebuttal letter, the authors write that they never referred their anyons as quasiparticles. But an anyon is referred to as an example of quasiparticle in their second sentence of the abstract.) Then they suddenly change the meaning of "anyon" to say that their object of interest is not a quasiparticle. The revision has made this subtlety more confusing to nonexperts, whom I believe the authors are trying to reach by publishing in a broadly read journal, Nature.

Let me reiterate why this should not be counted towards an experimental observation of nonabelian statistics. The authors have experimentally simulated nonabelian statistics.

Twist defects on a surface code simulate (projective) Ising anyon. The defect is not a quasiparticle of a persistent Hamiltonian system, so it is not a real anyon, but it captures essential elements of the anyon and imitates its behavior. That is what it means to simulate. The authors' experiment implements this simulation on a quantum device. Another aspect that distinguishes a simulation from an experimental verification is the degree of controllability. In simulation, one has reasonably complete control over all relevant degrees of freedom. A classical computer is an example since we can address each logical bit which is the only relevant degree of freedom. The present experiment is an example because the relevant degrees of freedom are qubits, not full microwave modes, and the device gives reasonably complete control over all those qubits. A fractional quantum Hall system provides an experimental verification, rather than a simulation, of a theory because it is simply impossible to address the many atoms and electrons in the condensed matter system using a few voltage/current probes and external magnetic field. If a trapped ion experiment realizes some special interaction on near-hundred atoms and observes some collective phenomena that was only speculated from the microscopic interaction by some crude approximate calculations which was best possible previously, then they can claim some experimental verification. Every experimental verification of a theoretical proposal contains such overwhelming ambiguity, and that is what makes the verification nontrivial. Therefore, the present experiment does not qualify for an experimental verification.

It appears to me that the authors want to say that a simulation is something nonphysical whereas their anyon is physical. The authors added a sentence "We therefore emphasize that the 2D braiding processes are physically taking place on the device, leading to actual non-Abelian exchange effects of local anyons in the manybody wavefunction, rather than matrix operations that simply follow the

same algebra." First, a minor problem here is that I don't think any reader can understand the reason or motivation of the last clause, "rather than matrix operations" without hearing my criticism on the previous version of the manuscript. Second, more importantly, the emphasis that the process is physical is rather absurd. Semiconductors would not quite work without quantum mechanics and my laptop would not give me an answer were it not for the manybody wavefunction of the electrons in my CPU. What is not physical in calculation on my laptop? It is an interpretation of physical states of electrons in the device. Isn't this experiment an interpretation of quantum states in the device?

Next, contrary to my criticism, the revision emphasized that this experiment "realizes key components of topological quantum computing." Let me expand my point why the authors should not claim anything about topological quantum computing. In short, topological approach to quantum computing is always tied to fault tolerance, about which this experiment says nothing. Given the public hype to the field, this matter is now an aspect of scientific integrity. Again, one should not reinvent or tweak the meaning of terms.

By topological quantum computing, we mean either, narrowly, a mechanism to realize a topologically ordered condensed matter system in which manipulation of quasiparticles implements quantum circuits, or, more broadly, a mechanism to realize fault-tolerant logical qubits and operations based on principles we learn from topological phases of matter. The narrow definition of topological quantum computing does not apply here as their system of interest is not governed by a persistent Hamiltonian with topological order. The broader definition could apply, only if it had elements of fault tolerance, but the authors do not attempt to do anything towards fault tolerance. If the current report is absolutely the best they can do (which I hope is not the case), I would conclude for myself that topological quantum computing using superconducting qubits is doomed because the best technology could not see diminishing logical error rates. This is precisely the opposite of what the authors want to say. If the authors can refute this criticism by data, the authors must do; if not, remove the claim.

To conclude, as I wrote in my first report, it is remarkable science to implement quantum simulation with such a high level of controllability, and this experiment is a good example in that regard. The problem is that the authors are selling their work for something their result does not qualify for. If the authors wish to clarify what their experiment is really doing, they can surely do, not by starting the discussion with highly sought-after anyons in the conventional sense, but by, for example, explaining what logical operations can be achieved by thinking about analogs(!) of anyons.

Referee #2 (Remarks to the Author):

The revised version of the manuscript by the Google group (now titled "Non-Abelian braiding of graph vertices in a superconducting processor") has acceptably addressed my initial concerns.

I am asked to give opinion whether the comments of the prior referees have been acceptably addressed. The referees certainly raise interesting points, some of which can be debated. The response of the authors, addresses these points in clear and productive ways. I believe the modifications to the manuscript which discuss these issues are good.

On referee 1 comments, while real matter and computer simulation of matter are clearly different, recent quantum experiments have made it hard to distinguish the two. We can argue what the dividing line is between if one actually has anyons and if one has a simulation of anyons. As the authors say, they have produced a wavefunction in the laboratory which is identical to the expected anyon wavefunction of the simulated system --- and they have checked some of its properties. This is not the same as simply simulating something on a classical computer or doing calculations analytically. So I disagree with referee 1 on this point, although I do agree that this is neither the same as having quasiparticles in a condensed matter system. So I think what is achieved is in between what referee 1 wants to claim and the ideal case. I also agree with referee 3 comments that this system is not topologically protected because there are no stabilizer measurements and error correction. The authors do acknowledge this point and simply leave the much more difficult objective for the future.

I think I would like to sit down with the other referees and debate what is and is-not achieved by this work. In the end I would think we would mostly agree. The key question is whether what has been done is interesting enough to warrant publication in a high profile journal. I feel that it is, but this is certainly subjective.

Referee #3 (Remarks to the Author):

1. I would like to thank the Authors for their detailed replies and extensive explanations. On the technical side I find that it is all crystal clear. In particular, I found the material added to the Supplementary quite helpful to understand the details of the work.

2. The clarity of the main text has also improved, thanks to the more extended discussion on topological protection and the different flavors of topological quantum computation. As a result I find that the nature of the main claims has been sharpened.

3. I also thank the Authors for their thoughts about the significance of this work for topological quantum computation. It is true that this is an important leap. I also think that the remaining leaps (adding quantum error correction) are arguably as important, and more difficult. The current breakthrough to topological quantum computation as a very long jump is to flying: we have verified that take-off is possible. However, I appreciate the significance attached to the realization of the non-Abelian exchange, even if it is not protected, and that the article will certainly generate a broad interest.

4. It has been appropriate to give more relevance to the theory works by Bombin and Kitaev, as pointed out by the second referee.

5. Finally, I find the main concern raised by the first referee quite important and not at all trivial. However, I think that it is to a good degree resolved by the new title chosen by the Authors, which I like very much:

"Non-Abelian braiding of graph vertices in a superconducting processor"

This new title emphasizes the (quantum) engineering nature of this work, as opposed to the previous title:

"Observation of non-Abelian exchange statistics on a superconducting processor"

(By the way, the Supplementary still uses this title; this should be fixed).

Let me elaborate on why this is important. By the use of the word "Observation", the previous title gave the impression that the Authors did not have fine control over every relevant degree of freedom of their device (i.e. the bosonic excitations of the superconducting circuit). Thus, it almost appeared as if the non-Abelian statistics were a phenomenon serendipitously observed in their processor by virtue of uncontrolled interactions happening in it (By the way, what a discovery that would be!).

Clearly this impression would be misleading, because I bet that the Authors, in view of their fine engineering and sound-proof pre-existing theoretical calculations, were happy but not really surprised to see non-Abelian exchange statistics show up at the end of their carefully prepared protocols.

Now, does this lack of surprise (related to the almost complete degree of control over the system) make the physical significance of the experiment as small as that of multiplying small matrices on my laptop, as the first referee suggests? I would disagree. I think the Authors are right in their reply. To explain why, let me indulge in an analogy. Consider these three scientific achievements:

- Team A discovers an alive mammoth in a remote corner of the world, and manages to domesticate it.
- Team B genetically engineers a docile mammoth starting from the genome of an elephant, and manages to bring it to life.
- Team C develops a virtual-reality environment that allows one to experience the encounter with a mammoth.

We would not call the achievement of Team B a simulation of a mammoth. Thus, it would be unfair to assimilate the achievements of Team B with those of Team C. It would also be unfair to assimilate the achievements of Team A and B, because they are of very different nature: one is a discovery, one an engineering breakthrough. The fact that it is genetically engineered does not make the mammoth of Team B less real than that of Team A.

Now let's make the analogy explicit.

- Team A discovers non-Abelian quasiparticles in the $5/2$ fractional quantum Hall state of a two-dimensional electron gas, and manages to braid them.
- Team B realizes the exchange of graph vertices in a superconducting processor and is able to verify that they obey non-Abelian statistics.
- Team C checks non-Abelian braiding statistics by performing a quantum algorithm for matrix multiplication on a system of 25 qubits.

The achievement of Team B is not a digital quantum simulation like that of Team C. Just like the two mammoths are both real, the phenomenon realized in the lab is as physically real as that observed by (hypothetical) Team A. This remains true even though Team B has almost complete control over their system while Team A has only very partial control and is otherwise "letting Nature do its work".

6. I think the Authors properly convey this distinction in the new crucial sentence added to the abstract. Here, I only object to the use of the term "microwave photons". I fear that the correct interpretation of the word "photon" in this context will only be appreciated by a small fraction of physicists, not to mention the broader readership. Thus I would avoid this jargon, or alas soon we will start hearing around that Google discovered that photons have non-Abelian statistics. I would recommend writing instead "the many-body wavefunction of the superconducting circuit" or something of the sort.

Pending this minor revision, and summing up all my remarks, my recommendation tilts towards publishing this work in Nature.

SIGNED: Bernard van Heck

Authors response to second round of review:

-Added the following sentence in the abstract in order to smoothen the transition from the challenges in realizing non-Abelian statistics to our approach:

“Controllable many-body quantum states generated on quantum processors offer another path for exploring these fundamental phenomena.”

-Changed “many-body wavefunction of microwave photons” to simply “many-body wavefunction” in the abstract

-Modified the last sentence of the abstract to emphasize that we have not implemented error correction and that our system is therefore not topologically protected. The sentence now reads:

“Our work provides new insights about non-Abelian braiding and- through the future inclusion of error correction to achieve topological protection- could open a path toward fault-tolerant quantum computing.”

-Rephrased the last sentence of the manuscript to make it more explicit that the inclusion of error correction is required to open a path toward topological quantum computing

-Updated the supplementary materials to reflect our title change.